# The Archaeal Proteome Project advances knowledge about archaeal cell biology through comprehensive proteomics

Stefan Schulze [1], Zachary Adams [2], Micaela Cerletti[3], Rosana De Castro [3], Sébastien Ferreira-Cerca [4], Christian Fufezan[5], María Inés Giménez[3], Michael Hippler [6,7], Zivojin Jevtic[8], Robert Knüppel[4], Georgio Legerme[1], Christof Lenz [8,9], Anita Marchfelder [10], Julie Maupin-Furlow [2,11], Roberto A. Paggi[3], Friedhelm Pfeiffer [12], Ansgar Poetsch[13,14,15], Henning Urlaub[8,9] & Mechthild Pohlschroder [1✉]

While many aspects of archaeal cell biology remain relatively unexplored, systems biology approaches like mass spectrometry (MS) based proteomics offer an opportunity for rapid advances. Unfortunately, the enormous amount of MS data generated often remains incompletely analyzed due to a lack of sophisticated bioinformatic tools and field-specific biological expertise for data interpretation. Here we present the initiation of the Archaeal Proteome Project (ArcPP), a community-based effort to comprehensively analyze archaeal proteomes. Starting with the model archaeon *Haloferax volcanii*, we reanalyze MS datasets from various strains and culture conditions. Optimized peptide spectrum matching, with strict control of false discovery rates, facilitates identifying > 72% of the reference proteome, with a median protein sequence coverage of 51%. These analyses, together with expert knowledge in diverse aspects of cell biology, provide meaningful insights into processes such as N-terminal protein maturation, *N*-glycosylation, and metabolism. Altogether, ArcPP serves as an invaluable blueprint for comprehensive prokaryotic proteomics.

[1] Department of Biology, University of Pennsylvania, Philadelphia, PA 19104, USA. [2] Department of Microbiology and Cell Science, Institute of Food and Agricultural Sciences, University of Florida, Gainesville, FL 32603, USA. [3] Institute of Biological Research (IIB-CONICET-UNMDP), National University of Mar del Plata, Mar del Plata 7600, Argentina. [4] Biochemistry III – Institute for Biochemistry, Genetics and Microbiology, University of Regensburg, 93053 Regensburg, Germany. [5] Institute of Pharmacy and Molecular Biotechnology, Heidelberg University, 69120 Heidelberg, Germany. [6] Institute of Biology and Biotechnology of Plants, University of Münster, 48143 Münster, Germany. [7] Institute of Plant Science and Resources, Okayama University, Kurashiki, Okayama 710-0046, Japan. [8] Bioanalytical Mass Spectrometry Group, Max Planck Institute for Biophysical Chemistry, 37077 Göttingen, Germany. [9] Institute of Clinical Chemistry, University Medical Center Göttingen, 37075 Göttingen, Germany. [10] Biology II, Ulm University, 89069 Ulm, Germany. [11] Genetics Institute, University of Florida, Gainesville, FL 32608, USA. [12] Computational Biology Group, Max Planck Institute of Biochemistry, 82152 Martinsried, Germany. [13] Plant Biochemistry, Ruhr University Bochum, 44801 Bochum, Germany. [14] Center for Marine and Molecular Biotechnology, Qingdao 266237, China. [15] College of Marine Life Sciences, Ocean University of China, Qingdao 266003, China. ✉email: pohlschr@sas.upenn.edu

Archaea are ubiquitous, play crucial roles in ecological processes, have impactful applications in biotechnology, and are more closely related to eukaryotes than are bacteria[1,2]. Yet, our understanding of archaeal cell biology is lacking behind eukaryotes and bacteria. Recently, the importance of proteomics as a tool for addressing specific biological questions in archaea has become readily apparent[3–13]. However, such limited analyses typically leave valuable information buried in the raw data. Fortunately, deposition of proteomic raw data in public repositories, such as PRIDE[14] or jPOST[15] is common practice. In the case of *Homo sapiens*, the Human Proteome Project (HPP) has demonstrated how the combination and reanalysis of proteomic datasets can lead to a more comprehensive map of the proteome, an improved genome annotation as well as substantial improvements in the understanding of biological and molecular functions[16–21]; however, comparable community efforts for prokaryotes have been lacking thus far.

While large-scale datasets for various prokaryotes exist, they are limited in their proteome coverage, analysis of various biological conditions, large-scale integration of multiple datasets and/or straightforward extensibility. The integration of multiple proteomics datasets for an archaeon was pioneered by the *Halobacterium salinarum* PeptideAtlas[22]. Despite the identification of 63% of the *H. salinarum* proteome, biological conclusions were scarce since only few culture conditions were analyzed and comparability between datasets was not given. Similarly, a Pacific Northwest National Laboratory library includes an impressive amount of bacterial and some archaeal proteomics raw files, but their analysis is mainly limited to peptide and protein identifications[23]. In regard to bacteria, large spectral libraries were generated, e.g. for *Staphylococcus aureus*[24,25] and *Mycobacterium tuberculosis*[26], with the latter being based on synthetic peptides, and facilitated the quantitative analysis of biomedically relevant samples. However, the application of spectral libraries is limited to similar instrumental setups and does not allow for discovery-driven approaches, which are crucial, e.g., for the analysis of post-translational modifications (PTMs). A concentrated effort of Schmidt et al. led to the development of *Escherichia coli* proteomics datasets that provided deep coverage of the proteome from different culture conditions[27]. But in all these examples, the combination of different datasets is largely missing, leading to a lack of comparisons between different strains and culture conditions. In addition, the extensibility of these collections is often not straightforward, as open-source analysis pipelines are not provided. Furthermore, the interdisciplinary expertise that is needed for the detailed analysis of proteomics datasets in regard to a multitude of biological questions, is enhanced through the involvement of research communities.

With the initiation of the ArcPP as a community project, we aim to shift prokaryotic proteomics toward a more comprehensive (re-)analysis of MS datasets. The ArcPP includes an increase in scale (by roughly an order of magnitude) of the combined datasets, extensive bioinformatic analysis of the detected proteins, the achieved depth of proteome sequence coverage as well as the comparison of datasets in regard to technical and biological aspects. Taken together, insights into archaeal cell biology are gained through this combined reanalysis of proteomic datasets, supported by interdisciplinary expertise.

## Results and discussion

**Optimized large-scale reanalysis of diverse datasets**. *H. volcanii* is a halophilic archaeon and, facilitated by a wide range of genetic and molecular biology tools[28], it is the model of choice to study a variety of cellular processes, leading to the most extensive proteomic studies completed amongst archaea thus far

(Supplementary Table 1). Therefore, we chose to perform our initial reanalysis on 12 diverse *H. volcanii* MS datasets comprising more than 23 million spectra (Fig. 1). These reanalyses facilitated not only a deep coverage of the proteome but also revealed differential protein identification dependent on culture conditions, as we show here. In addition, differences in protein digestion, peptide fractionation and MS measurements enabled comparisons regarding optimal sample processing. Notably, various datasets used different quantitative approaches, allowing for the future integration of protein dynamics across multiple experiments.

For the unified, large-scale analysis of all datasets, we used the Python framework Ursgal[29]. Key aspects of this reanalysis include: (i) an initial optimization of search parameters like precursor and fragment ion mass tolerances, (ii) the use of the most recent protein database derived from an updated genome annotation, and (iii) the use of three protein database search engines. In addition, the use of multiple search engines allowed to apply a combined posterior error probability (PEP) approach[29,30], which rescores peptide spectrum matches (PSMs) based on their overlap between the different search engines, thereby taking advantage of an increased confidence in shared PSMs. Each of these steps aimed to increase the number of correct PSMs while at the same time reducing the number of false positives. A comparison of the results from this reanalysis to the original search results showed for six datasets an increased number of PSMs and/or identified peptide sequences by more than 10%, while for only three datasets a slight decrease in identifications was noted (Fig. 2a). Decreases could be attributed to peculiarities in the experimental setup or analysis details of these datasets (Supplementary Note 1). The optimization of search parameters and the combined PEP approach demonstrated their usefulness in all cases (exemplified in Fig. 2a, bottom). Importantly, these results were achieved while tightly controlling the PEP (≤1%), which is a more conservative approach to error rate control than is the use of false discovery rates (FDRs)[31]. Therefore, this approach provided a unified and optimized large-scale analysis of all available *H. volcanii* datasets.

**Combining datasets for increased proteome coverage**. When aggregating results from multiple large datasets, FDRs must be controlled on both the peptide and protein level to avoid the accumulation of false positives as the overall dataset size increases[21,32]. We monitored FDR distributions for peptides as well as proteins and used recently established approaches to ensure identifications with high confidence. For peptides, we observed a bias toward higher FDRs for small (<10 amino acids) and large peptides (Fig. 2b, for peptide length distribution see Supplementary Fig. 1a). Therefore, we adopted the approach used by the MassIVE Knowledge Base[21] to calculate FDRs for groups of peptides with the same lengths. On the protein level, a picked protein FDR approach was applied, which calculates FDRs based on a comparison of targets with their corresponding decoys. This approach had been shown to be applicable to large datasets and provides a more accurate FDR estimation[32]. When applied to the ArcPP, this strategy resulted in a better separation between targets and decoys, and even allowed to increase analysis stringency by reducing the FDR threshold to 0.5% instead of the common 1% without decreasing the number of identified proteins substantially (Fig. 2c). Finally, the identification of a peptide sequence or protein was considered highly confident only if it was based on a minimum of two spectra, further improving separation between targets and decoys especially on the peptide level (Supplementary Fig. 1b, c).

Using these strict criteria, a total of 40,877 peptide sequences corresponding to 2930 proteins were identified (Fig. 3a),

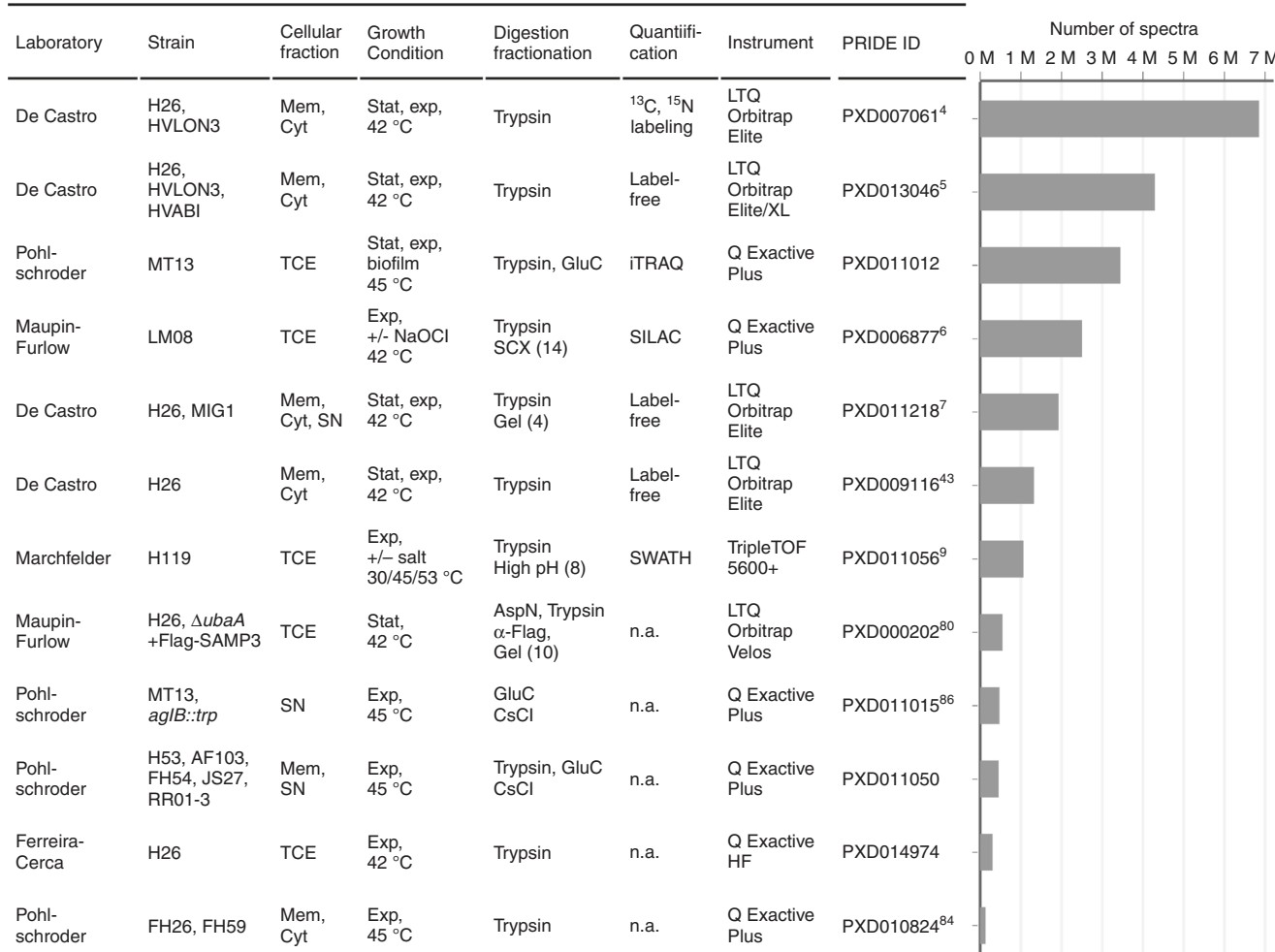

| Laboratory | Strain | Cellular fraction | Growth Condition | Digestion fractionation | Quantification | Instrument | PRIDE ID | Number of spectra |
|---|---|---|---|---|---|---|---|---|
| De Castro | H26, HVLON3 | Mem, Cyt | Stat, exp, 42 °C | Trypsin | $^{13}C$, $^{15}N$ labeling | LTQ Orbitrap Elite | PXD007061[4] | |
| De Castro | H26, HVLON3, HVABI | Mem, Cyt | Stat, exp, 42 °C | Trypsin | Label-free | LTQ Orbitrap Elite/XL | PXD013046[5] | |
| Pohl-schroder | MT13 | TCE | Stat, exp, biofilm 45 °C | Trypsin, GluC | iTRAQ | Q Exactive Plus | PXD011012 | |
| Maupin-Furlow | LM08 | TCE | Exp, +/- NaOCl 42 °C | Trypsin SCX (14) | SILAC | Q Exactive Plus | PXD006877[6] | |
| De Castro | H26, MIG1 | Mem, Cyt, SN | Stat, exp, 42 °C | Trypsin Gel (4) | Label-free | LTQ Orbitrap Elite | PXD011218[7] | |
| De Castro | H26 | Mem, Cyt | Stat, exp, 42 °C | Trypsin | Label-free | LTQ Orbitrap Elite | PXD009116[43] | |
| Marchfelder | H119 | TCE | Exp, +/- salt 30/45/53 °C | Trypsin High pH (8) | SWATH | TripleTOF 5600+ | PXD011056[9] | |
| Maupin-Furlow | H26, ΔubaA +Flag-SAMP3 | TCE | Stat, 42 °C | AspN, Trypsin α-Flag, Gel (10) | n.a. | LTQ Orbitrap Velos | PXD000202[80] | |
| Pohl-schroder | MT13, aglB::trp | SN | Exp, 45 °C | GluC CsCl | n.a. | Q Exactive Plus | PXD011015[86] | |
| Pohl-schroder | H53, AF103, FH54, JS27, RR01-3 | Mem, SN | Exp, 45 °C | Trypsin, GluC CsCl | n.a. | Q Exactive Plus | PXD011050 | |
| Ferreira-Cerca | H26 | TCE | Exp, 42 °C | Trypsin | n.a. | Q Exactive HF | PXD014974 | |
| Pohl-schroder | FH26, FH59 | Mem, Cyt | Exp, 45 °C | Trypsin | n.a. | Q Exactive Plus | PXD010824[84] | |

**Fig. 1 Summary of ArcPP datasets comprising a total of more than 23 million spectra.** A diverse array of MS datasets for *H. volcanii* has been compiled for the initial reanalysis by the ArcPP. For each dataset, strains (separated by comma), cellular fractions (Mem, membrane; Cyt, cytosol; SN, culture supernatant, TCE, total cell extract), growth conditions (stat, stationary; exp, exponential growth phase), enzyme(s) used for protein digestion, and fractionation methods on peptide (SCX, strong cation exchange chromatography; high pH, high pH reversed-phase chromatography) or protein level (gel, SDS-PAGE; CsCl, CsCl gradient) with the number of fractions indicated in parentheses, quantification methods (iTRAQ, isobaric tags for relative and absolute quantitation; SILAC, stable isotope labeling with amino acids in cell culture; SWATH, sequential window acquisition of all theoretical fragment ion spectra), instruments employed, corresponding PRIDE IDs with references and the sum of all spectra are noted. Experiments were performed by five different laboratories. For more details see Supplementary Tables 2-4 and Supplementary Data 1-2. Source data are provided as a Source data file.

representing 72% of the predicted 4074 proteins encoded by the *H. volcanii* genome (45,533 peptide sequences and 3010 proteins if identifications based on a single PSM and FDR ≤ 1% were included, Supplementary Fig. 1d). Furthermore, the high number of identified peptides also resulted in a remarkably high median protein sequence coverage of 51% (Fig. 3b). This coverage is the most comprehensive draft of an archaeal proteome achieved thus far, and this work illustrates the value of combining multiple datasets, as the identifications and sequence coverage resulting from this reanalysis greatly exceed the numbers for each individual dataset.

**Comparison of MS sample processing approaches.** By considering the number of confident identifications in light of the different experimental setups, one can draw conclusions about sample processing and MS methods, which in turn can improve the design of future experiments. While technical aspects are discussed in more detail in Supplementary Note 2, we want to highlight some key findings here. As expected, identification rates

were mainly dependent on the resolution and sensitivity of the instrument (Supplementary Fig. 2). Interestingly the use of peptide fractionation (PXD006877 and PXD011056) resulted in the highest number of protein identifications, while the most peptide identifications were obtained by using multiple, complementary proteases (trypsin and GluC), even without fractionation (PXD011012). Furthermore, by analyzing the characteristics of identified and missing proteins, we revealed a strong decrease in identification rates for proteins <13 kDa (Supplementary Fig. 3a). This highlights that although small proteins recently gained attention[33–35], their identification still requires major improvements. Similarly, the identification of integral membrane proteins is generally challenging[36]. Here the identifications for hydrophobic proteins (grand average of hydrophobicity (GRAVY) > 0, Supplementary Fig. 3c) was less than for non-hydrophobic proteins with solubilization by SDS showing a remarkable improvement over, e.g., TRIzol extraction (Supplementary Fig. 4a) for hydrophobic protein identification. In total, 55% of predicted integral membrane proteins were identified (Fig. 3c).

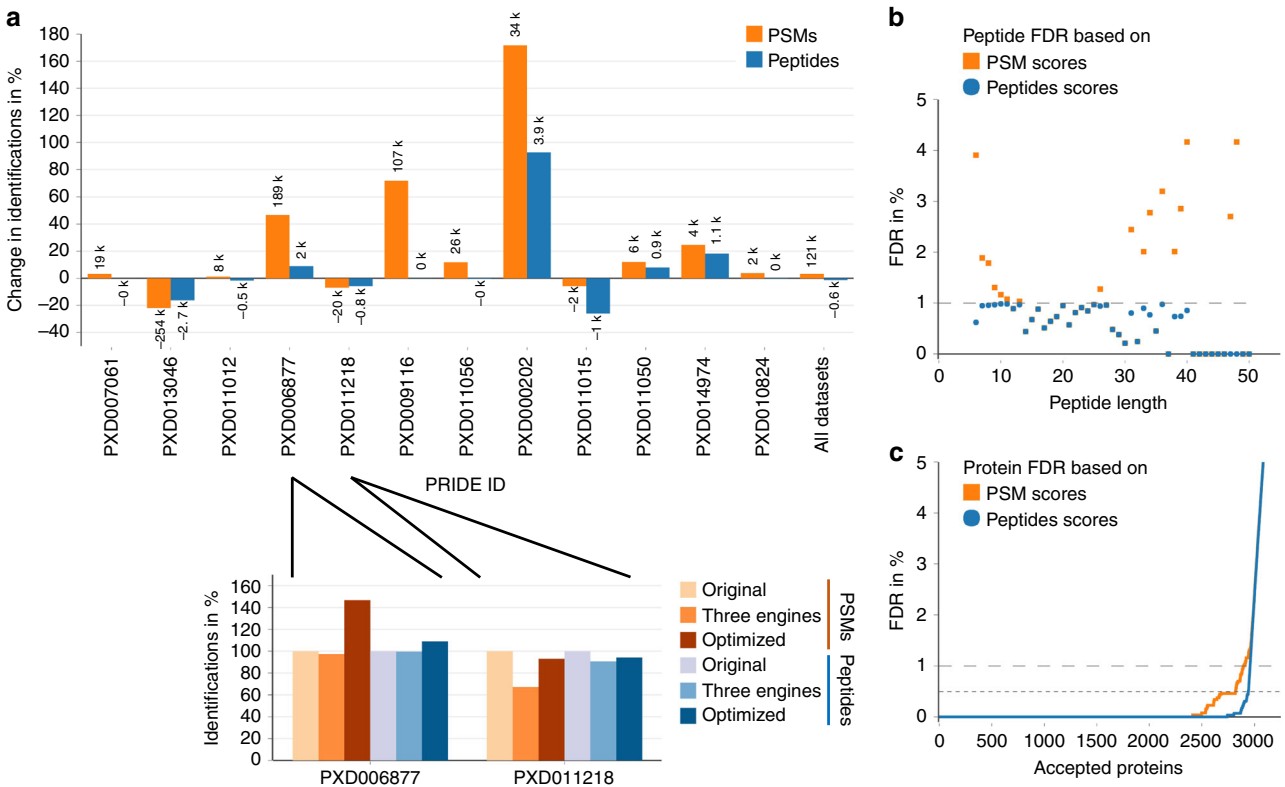

**Fig. 2 Optimized reanalysis of datasets and strict control of FDRs. a** A unified dataset reanalysis was performed with Ursgal[29], including search parameter optimization (parameter sweep iterating through all combinations of a set of four different precursor mass tolerances, four fragment mass tolerances and ten instrument offsets) as well as a combination of three protein database search engines. Results were compared with the original identifications reported for each dataset and differences are given for the number of PSMs (orange) and identified peptide sequences (blue) on a percentage basis (height of the bar, 0% corresponds to the original results) and as absolute numbers (indicated above/below each bar) for each dataset. For two exemplary datasets, the effects of using three protein database search engines and an optimized reanalysis, including optimization of search parameters, as well as the combined PEP approach[29,30], are shown in comparison to the original results (normalized to 100%) in the bottom panel. **b** For each peptide length, the FDR for all peptide sequences within this group was determined after (i) including all PSMs with a PEP ≤ 1% (orange) and (ii) adjusting the FDR on peptide level (blue). **c** Protein FDRs are shown for the number of accepted proteins (ranked by protein q-value) after (i) including all PSMs with a PEP ≤ 1% (orange) and (ii) adjusting the FDR on protein level using the picked protein FDR approach[31] (blue). It should be noted that filtering for identifications based on at least two PSMs removed all decoy hits on the peptide level (resulting in a theoretical FDR of 0%), while it did not substantially affect the target-decoy distribution on protein level (Fig. S1). Source data are provided as a Source data file.

While this is still lagging behind the identification rates for cytosolic proteins (> 75%), it is nevertheless a notable improvement over previous studies for this challenging subproteome[7,37,38].

**N-terminal protein processing and cell surface homeostasis.** Furthermore, the high protein sequence coverage achieved within the ArcPP allowed for the large-scale analysis of N-terminal protein maturation in *H. volcanii*. The identification of 1085 N-terminal peptides for 27% of all predicted proteins represents a more than 6-fold increase compared with previous studies[39,40] and is even higher than the identification rate in a recent, dedicated approach for *Sulfolobus islandicus*[10]. Our data confirm that cleavage of methionine occurs for the majority of proteins and that N-terminal acetylation of cleaved and uncleaved termini is common in *H. volcanii* (Fig. 3d)[39,40]. With the identification of a broader range of substrates, ArcPP results suggest that N-terminal protein maturation takes place similarly for cytosolic and integral membrane proteins. Interestingly, while acetylation of uncleaved methionine was reported for *H. volcanii*[40] as well as the evolutionary distant *S. islandicus*[10] and *S. solfataricus*[41], it was not detected in the closely related *H. salinarum* and

*Natronomonas pharaonis*[42]. A reanalysis of *Natrialba magadii* proteomics data (PXD009116[43]) revealed acetylation of uncleaved methionine as well. Taking this into account, the GCN5-related N-acetyltransferases (GNAT) domain containing HVO_2604 is a candidate for catalyzing the N-acetylation of methionine in *H. volcanii* as it lacks an ortholog in *H. salinarum* and *N. pharaonis*, but has an ortholog in *N. magadii* (Nmag_1976). Furthermore, HVO_2604 is encoded adjacent to the signal peptidase gene (sec11a, HVO_2603) and methionine aminopeptidase (HVO_2600) homologs in *H. volcanii* (but not in *N. magadii*). Alternative GNAT candidates include *H. volcanii* HVO_1954 and its *N. magadii* ortholog (Nmag_1596) as they share 3D-structural homology and conserved active site residues with the *S. solfataricus* SsArd1 shown to catalyze the N-acetylation of diverse protein substrates including those with methionine N-termini[10,41,44]. We note that the deletion of *SsArd1* in *S. islandicus* was shown to lead to growth defects[10] while alterations in N-acetylation of the 20S proteasomal alpha1 protein in *H. volcanii* affected growth and stress tolerance[40], both demonstrating the importance of N-terminal acetylation. The identification of a broad range of substrates within the ArcPP as well as GNAT candidates now allow elucidating the cellular functions of this modification in more detail.

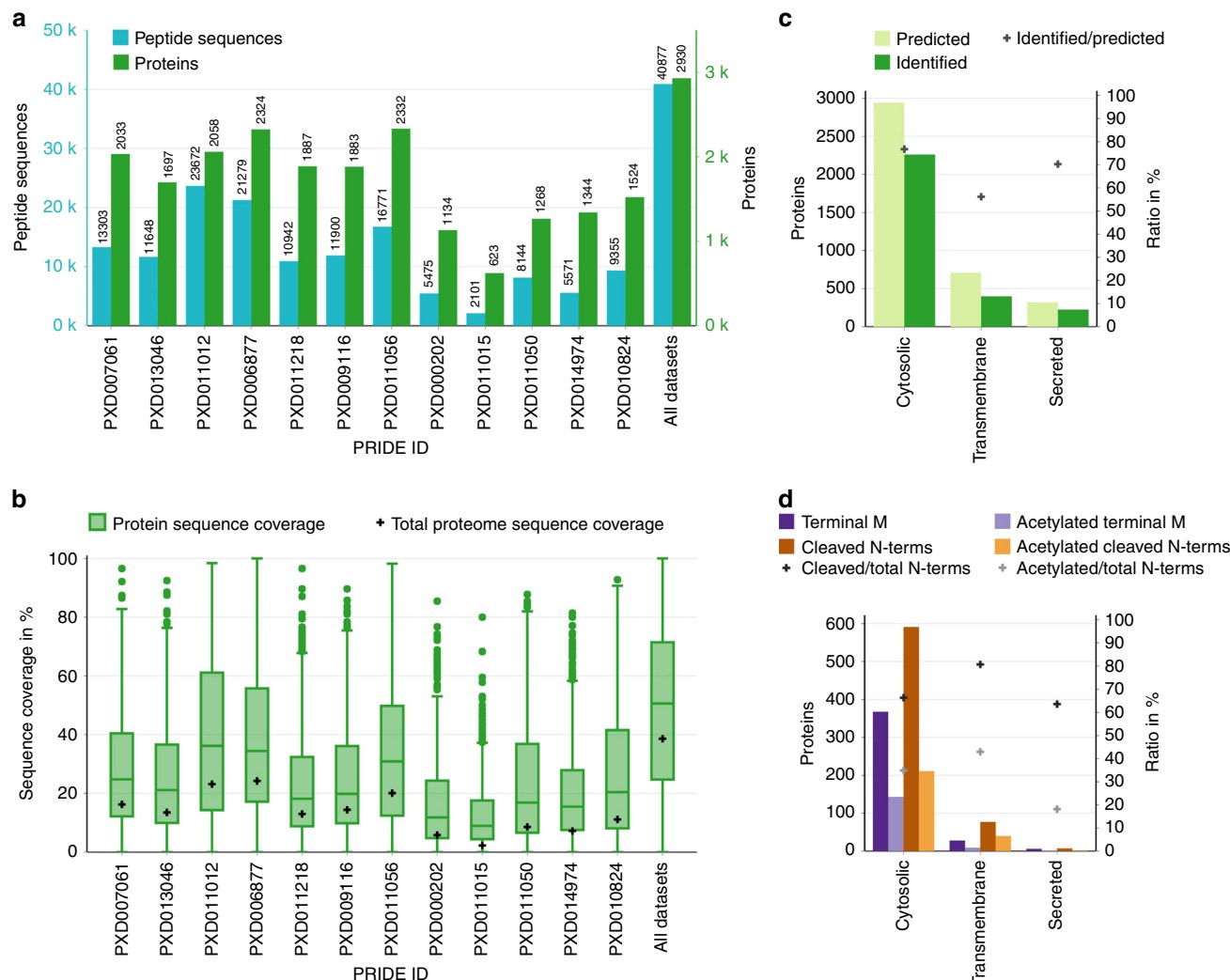

**Fig. 3 Highly confident identification of 2930 proteins with a median sequence coverage of 51%. a** The number of identified peptide sequences (blue) and proteins (green), with a peptide FDR ≤ 1% and protein FDR ≤ 0.5%, respectively, as well as at least two PSMs, is given for each dataset as well as the combination of all datasets. **b** Box-plots for the sequence coverage of all confidently identified proteins (green, number of proteins for each dataset is given in Fig. 3a) as well as the total proteome sequence coverage (black crosses) is given for each dataset and the combination of all datasets (center line, median; box limits, upper and lower quartiles; whiskers, 1.5x interquartile range; points, outliers). **c** All *H. volcanii* proteins were grouped into localization categories based on the integration of multiple prediction engines. The number of predicted (light green) and identified (dark green) proteins as well as the identification rate (cross) is given (for more detailed data see Supplementary Fig. 4). **d** For these categories, based on 1085 identified N-terminal peptides, the N-terminal protein maturation has been analyzed. The number of proteins with terminal methionine (dark purple) or cleaved N-terminus (up to one amino acid, dark orange) as well as their acetylated counterparts (light purple and light orange, respectively) are given. In addition, the ratio of cleaved and acetylated N-termini (black and gray cross, respectively) to the total number of identified N-termini in each category is indicated. Note that one protein can be identified with different N-terminal peptidoforms and would be counted for each corresponding category. Source data are provided as a Source data file.

In addition to cytosolic and integral membrane proteins, 70% of the proteins predicted to be transported across the membrane and N-terminally processed by different secretion pathways were identified (Fig. 3c and Supplementary Fig. 4b). Notably, 1045 C-termini were identified in total covering a large percentage of these secreted proteins (Supplementary Fig. 4c), but almost none of the N-termini of the secreted proteins were detected (Supplementary Fig. 4d). These data suggest the presence of signal peptides, supporting the results of the corresponding prediction engines. However, these programs thus far are trained on a very limited number of experimentally verified archaeal processing sites[45–48]. Therefore, taking advantage of the extensive data available within the ArcPP, semi-enzymatic database searches were performed for datasets that

used trypsin for proteolytic digestion. Results were compared with signal peptide cleavage sites (CS) predicted by SignalP 5.0[48]. For 11 and two substrates of the Sec and Tat pathway, respectively, the predicted signal peptidase I (SPI) CS could be confirmed (Supplementary Fig. 4e). In addition, for three and one protein(s) of the same secretory pathways, respectively, the CS could be refined. This approximately doubles the number of confirmed processing sites identified for archaea so far (Supplementary Fig. 4f). Together with proteins, for which fully enzymatic peptides show evidence of a false positive signal peptide prediction (Supplementary Note 3), these results will allow for the optimization of archaeal prediction programs and hence improve the identification of protein processing and subcellular localization. This finding is invaluable for an

improved understanding of archaeal cell surface biogenesis, a crucial aspect for the interaction of archaea with their environment.

Another important aspect of cell surface homeostasis are membrane-associated proteases like LonB and rhomboid protease RhoII. The former is involved in the regulation of cell shape and carotenoid biosynthesis in *H. volcanii* while a knockout of the latter affected the *N*-glycosylation of the S-layer glycoprotein with a sulfoquinovose-containing oligosaccharide[4,5,7,13,49]. The datasets PXD007061/PXD013046 and PXD011218 originally characterized the proteomes of a conditional LonB mutant and a RhoII knockout mutant, respectively. The reanalysis of these datasets within the ArcPP has now identified four additional integral membrane proteins as probable RhoII targets as well as three previously undescribed potential LonB substrates (Supplementary Note 4), which can help to deepen our understanding of the biological roles of RhoII and LonB, respectively.

**Proteins identified across a variety of growth conditions.** In order to gain further insights into cell biological aspects in archaea, we focused on the comparison of datasets with regard to commonly or uniquely identified proteins. Seven of the datasets used in our reanalysis (PXD007061, PXD013046, PXD011012, PXD006877, PXD011218, PXD009116, and PXD011056), comprising 2912 protein identifications, were suitable for such a comparison since they analyzed either total cell extracts or a combination of membrane and cytosolic fractions and can therefore be regarded as covering the complete proteome. Approximately half of the proteins are included in at least six of the seven datasets (Fig. 4a), indicating that these proteins have crucial functions under vastly distinct conditions. In line with this, out of 60 genes that are considered essential, because corresponding deletion mutants could not be generated in *H. volcanii* so far (Thorsten Allers and the Haloferax community, personal communication), 47 were identified in at least six datasets. This includes translation initiation factors (Ttif1a, Tif2c)[50], the membrane-associated LonB protease[51] and secretory pathway proteins such as SRP54[52] and TatCt[53]. Similarly, more than 80% of homologs to essential genes identified by transposon tagging (TnSeq data) in *S. islandicus*[54] (excluding small proteins <15 kDa) were detected in most whole-cell datasets. In contrast to genetic analyses, the proteomic approach presented here can also indicate crucial functions of proteins for which corresponding individual genes are dispensable. For example, thermosome (Ths1/2/3) and proteasome (PsmA1/2) components could be deleted individually but not altogether, while PsmB, another proteasome component, was demonstrated essential based on a conditional lethal mutation;[55,56] these proteins were identified in at least six datasets. Our findings are also consistent with an enrichment of arCOG classes[57] representing core physiological functions like translation or nucleotide and energy metabolism (Fig. 4b and Supplementary Fig. 5), which had been shown to contain high numbers of essential genes[54].

Also present in all datasets were the highly abundant S-layer glycoprotein, the sole subunit of the *H. volcanii* cell envelope, and nearly all known components of the two known *H. volcanii* *N*-glycosylation pathways (AglB- and Agl15-dependent pathways, Fig. 4b)[58], illustrating the importance of *N*-glycosylation in *H. volcanii*. Notably, however, the Agl15-dependent *N*-glycosylation pathway was proposed to be active only under low salt conditions[59,60]. Our metaproteomic finding raises the question as to whether Agl15-dependent *N*-glycosylation occurs under additional culture conditions or is regulated in activity posttranslationally. Interestingly, both the membrane proteases RhoII and LonB, which were identified in all whole proteome datasets,

are thought to be implicated in the regulation of the protein glycosylation process in *H. volcanii*[4,49].

**Protein identifications unique to specific growth conditions.** Conversely, identification of proteins in only one dataset can provide valuable insights into the possible functions of these proteins, such as roles in acclimation to specific stresses or in regulatory processes. Differences in sample processing and MS acquisition techniques can also influence protein identification between datasets. However, the frequent detection of proteins with common physicochemical properties (see above) or even multiple proteins of the same pathway within a distinct dataset strongly suggests that they play important roles under specific conditions. For example, multiple subunits of urease (UreA, UreB) and associated maturation proteins (UreE, UreF) were only detected in the dataset PXD006877, the only dataset in which glycerol minimal medium (GMM) was used. Ureases are important in nitrogen cycles including the conversion of fertilizers to ammonia gas, yet, urease activity was suggested to be rare in halophiles[61,62]. This presumed restriction in activity is in contrast to predicting urease gene homologs in many haloarchaea in operons similar to those of the *Thaumarchaeota* (Supplementary Fig. 6A, B), for which urease activity is widely distributed[63,64]. Within the ArcPP, UreE and UreF, important for Ni$^{2+}$ insertion into the urease active site, were only identified in GMM. Together with the increased transcription of corresponding genes in GMM[65], this suggests that urease expression in halophiles is linked to specific environmental conditions including carbon sources. To test this hypothesis, a phenol-hypochlorite method, compatible with hypersaline conditions, was used to assay the catalytic generation of ammonia from urea (Supplementary Fig. 6C). This approach showed the hydrolysis of urea in cell lysates of *H. volcanii* grown to log phase in GMM (Supplementary Fig. 6D). The temperature optimum was determined to be around 60 °C, which is 15 °C above the growth temperature optimum of *H. volcanii*[66] and similar to the temperature optimum of the urease activity detected in *Haloarcula hispanica*[62]. In contrast, the urease activity of *H. volcanii* cells grown on complex media (CM) was undetectable (Supplementary Fig. 6D). This finding indicates that the mass spectrometrically detectable presence of urease subunits is indeed correlated with urease activity and regulated by metabolic status. These findings have implications for determining urea turnover in hypersaline environments.

Regarding the biosynthesis of type IV pili, the ATPase PilB3 was reliably identified in all total proteome datasets. This is consistent with an inability of *H. volcanii* to form detectable pili and a significant reduction of surface adhesion when *pilB3* and *pilC3* are deleted[67]. However, *H. volcanii* contains five *pil* operons and encodes multiple pilins and their biological roles are yet largely unknown[67,68]. Interestingly, PilB1 and PilB4 are only found in cells grown with GMM, thereby providing experimental conditions to study the roles of these PilB paralogs and their corresponding pili.

Finally, the majority of non-identified proteins (69%) has physicochemical properties (small, alkaline, hydrophobic) associated with reduced identification rates. However, we detected four genomic islands with a low protein identification rate, among them two predicted proviruses (Supplementary Note 5). Another region with mostly lacking protein identifications (HVO_B0160 to HVO_B0181) includes genes linked to respiratory nitrate reductase (HVO_B0161 HVO_B0166) which is only transcribed under anaerobic conditions[69]. This finding highlights that *H. volcanii* has not been proteomically analyzed under anaerobic conditions so far and hints at further proteins that might play a role in the response to anoxia.

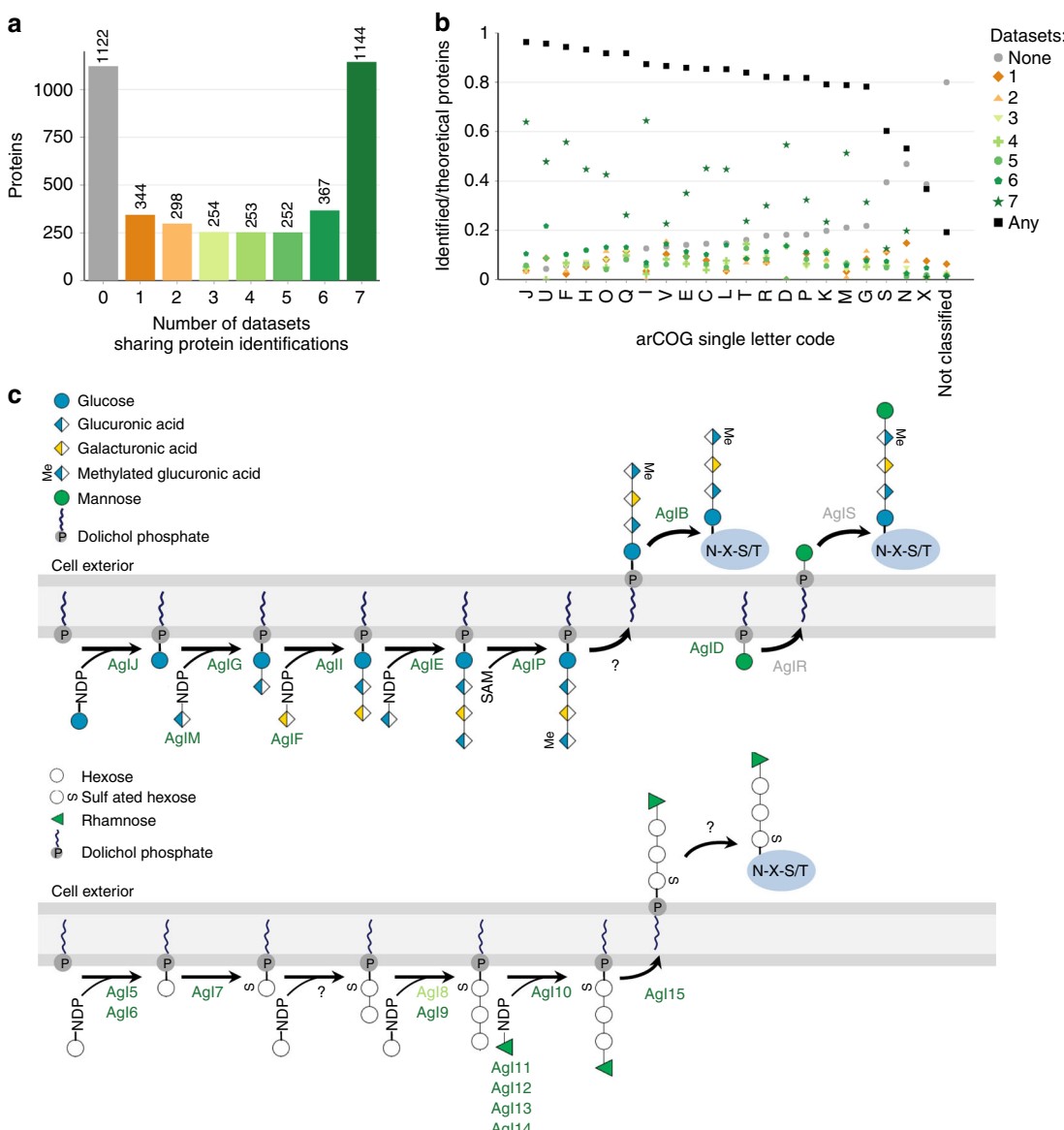

**Fig. 4 Comparison of whole proteome datasets revealing ubiquitous presence of *N*-glycosylation pathway enzymes. a** The overlap in protein identifications between seven datasets that analyzed samples of whole proteomes has been determined. The number of proteins identified in the given number of datasets (1–7, green to orange, throughout this figure) is represented as a bar plot. For proteins that were not identified (0, gray), all ArcPP datasets were taken into account. **b** For each arCOG class (sorted by ArcPP identification rate), the identification rate for all proteins belonging to that class is given based on all proteins identified within the ArcPP (black), proteins not detected within the ArcPP, and proteins that are part of 1–7 whole proteome datasets. ArCOG classes are as follows: J, translation, ribosomal structure and biogenesis; U, intracellular trafficking, secretion, and vesicular transport; F, nucleotide transport and metabolism; H, coenzyme transport and metabolism; O, post-translational modification, protein turnover, chaperones; Q, secondary metabolites biosynthesis, transport and catabolism; I, lipid transport and metabolism; V, defense mechanisms; E, amino acid transport and metabolism; C, energy production and conversion; L, replication, recombination and repair; T, signal transduction mechanisms; R, general function prediction only; D, cell cycle control, cell division, chromosome partitioning; P, inorganic ion transport and metabolism; K, transcription; M, cell wall/ membrane/envelope biogenesis; G, carbohydrate transport and metabolism; S, function unknown; N, cell motility; X, mobilome. **c** The known steps of the two described *N*-glycosylation pathways in *H. volcanii* (top and bottom, AglB- and Agl15-dependent, respectively, based on refs. [58,74,113]) are schematically shown with their corresponding enzymes colored according to the number of datasets in which they have been identified. Interestingly, while almost all known enzymes were identified in at least six datasets, AglR and AglS were not identified in these datasets at all. Notably, these are involved in the addition of the final mannose to the AglB-dependent glycan and *N*-glycopeptides with and without the final mannose attached have been readily identified previously[86,114]. Source data are provided as a Source data file.

**Enabling further insights and community contributions**. While these examples give early indications of how important information can be harvested from peptide and protein identifications, naturally, quantitative analyses of suitable datasets within the ArcPP will eventually lead to even deeper insights into the

mechanisms underlying specific regulatory processes and stress responses in *H. volcanii*. At the same time, increased efforts are required to unravel the function of large parts of archaeal proteomes[70], since 15% of proteins present in all whole-cell datasets are of unknown function and even 40% of proteins unique to one

dataset (Supplementary Fig. 5). Moreover, the exceptionally high protein sequence coverage achieved here enables proteogenomic analyses that will lead to an improved genome annotation[71,72]. We already identified eight proteins that were annotated as nonfunctional, providing evidence for the existence of these proteins (Supplementary Note 6). Similarly, ArcPP is ideal for the validation of gene models based on transcriptomics and ribosome profiling data[73]. Finally, given the low abundance of many types of PTMs, high protein sequence coverage is essential to the identification of peptides decorated by these. While the presence of some PTMs has been confirmed in *H. volcanii*[39,74–76] and other archaea[3,77], comprehensive analyses are still lacking.

In conclusion, we have illustrated that the reanalyses performed by the ArcPP have proven suitable for providing valuable insights into archaeal cell biology. Furthermore, the ArcPP allows for informed decisions about approaches to answer emerging biological questions. Since this resource provides invaluable information for the archaeal community, we have made our results available through a searchable web database at https://archaealproteomeproject.org. In addition, the most recent, annotated *H. volcanii* protein database, the meta data for all experimental datasets and summary files for all highly confident identifications on PSM, peptide and protein level are accessible at https://github.com/arcpp/ArcPP. Since the number of proteomic datasets available for *H. volcanii* continues to grow, analysis scripts are provided that will facilitate a straightforward reproduction and extension of results, which can be easily contributed and integrated into the ArcPP through GitHub. This workflow is especially important for the community-driven extension of this approach toward other archaeal species, for many of which large-scale proteomics datasets already exist (Supplementary Table 1). Finally, the ArcPP can serve as a blueprint for comprehensive bacterial proteomics with even greater availability of public datasets.

## Methods
**Datasets collected for *H. volcanii*.** All datasets reanalyzed here were originally uploaded to PRIDE[14] through ProteomeXchange[78] or jPOST[15] and are accessible via their corresponding PRIDE ID (Supplementary Data 3). Details about the analyzed strains, experimental conditions, MS instruments and settings can be found in Supplementary Note 7, Supplementary Table 2, Supplementary Data 1-2 as well as the corresponding publications (if available). Therefore, only a short summary of each dataset will be given here. The following datasets are considered analyses of whole proteomes: PXD006877, PXD007061, PXD009116, PXD011012, PXD011056, PXD011218, and PXD013046. For reference, the theoretical proteomes that had been exported from HaloLex[79] and were used in some of the previous analyses have now been made available via Zenodo, together with the proteome that was used for all analyses within the ArcPP. The set of proteomes is available at https://doi.org/10.5281/zenodo.3565580. It should be noted that all used strains are direct descendants of the type strain DS2, which was used for genome sequencing and thus for the reference proteome.

**Dataset PXD000202.** This dataset has been previously published by Miranda et al.[80]. Sampylation is a mechanism of ubiquitin-like protein modification in Archaea[76]. *H. volcanii* encodes three ubiquitin-like small archaeal modifier proteins (SAMP1-3) that are covalently attached to target proteins by a mechanism that requires the E1-like activating enzyme UbaA[80,81]. To map the sites of samp3ylation, in which the SAMP3 C-terminal Gly is covalently linked to the ε-amino group of lysine residues of target proteins, the following strategy was used. Samp3ylated proteins were purified by α-Flag chromatography from cells expressing SAMP3 with an N-terminal Flag-tag. To improve the MS-based mapping of samp3ylation sites, the alanine residue (Ala90) immediately N-terminal to the diglycine motif of SAMP3 was modified to a lysine residue (A90K). This amino acid exchange allowed for scanning for GG-footprints derived from SAMP3 on tryptic peptides of the target protein by detecting mass increases of +114 Da. Samp3ylated proteins were purified from wild-type and compared with an isogenic E1 mutant (ΔubaA) deficient in the ability to activate the SAMPs for ubiquitin-like modification or sulfur mobilization. This latter strain enabled us to establish that the sites identified by MS analysis were dependent upon the E1 enzyme. *H. volcanii* strains were grown to stationary phase in ATCC974 complex medium (200-ml cultures). Clarified cell lysates were applied to equilibrated α-Flag columns, washed, and eluted with 100 μg ml⁻¹ 1X Flag peptide and collected in nine fractions. Wild-type and ΔubaA mutant strains expressing Flag-SAMP3A90K were analyzed in biological triplicate and duplicate, respectively. Proteins purified by α-Flag chromatography were separated by 15% nonreducing SDS-PAGE. Each gel lane was cut into 10 gel pieces and digested with trypsin. Peptide fragments were subjected to reversed-phase column chromatography operated on an Easy-nLC II connected to an LTQ Orbitrap-Velos mass spectrometer. Acquired MS/MS spectra were originally searched against a Uniprot *H. volcanii* DS2 proteome using the Sorcerer-SEQUEST platform[82]. Cysteine carbamidomethylation, methionine oxidation, and diglycyl-lysine were set as variable modifications.

**Dataset PXD006877.** This dataset has been previously published by McMillan et al.[6]. Multiplex quantitative stable isotope labeling in cell culture (SILAC) was used to monitor the changes in the *H. volcanii* proteome during hypochlorite stress. A double auxotroph for lysine and arginine (LM08) was generated and used to fully incorporate the heavy amino acids ¹³C/¹⁵N-lysine (+8 Da) and ¹³C-arginine (+6 Da) into each peptide. Cells were grown in GMM supplemented with the heavy vs. light amino acids (0.3 mM each). At late-log phase, cells were treated for 20 min with the oxidizing agent (2.5 mM NaOCl) vs. a mock (ddH₂O) control. After treatment, harvested cell pellets of control and treatment groups were mixed at a 1:1 ratio ($n = 4$ biological replicates with a label swap). Proteins were extracted with TRIzol and solubilized in buffer (7 M urea, 2 M thiourea and 4% (w/v) CHAPS). After reduction with tris-(2-carboxyethyl) phosphine (TCEP), cysteine side chains were blocked by methyl methanethiosulfonate (MMTS) treatment. Digestion with trypsin was followed by desalting on a C18 reverse phase mini-column. Eluted peptides were lyophilized and fractionated into 14 fractions by strong cation exchange chromatography (SCX). SCX fractions were analyzed one at a time on an Easy-nLC 1200 system coupled to a Q Exactive Plus mass spectrometer. The original peptide identification and quantification was performed with Proteome Discoverer 2.1 using the Uniprot *H. volcanii* DS2 proteome. Methylthio was included as fixed modification and lysine + 8, arginine + 6, proline + 5, methionine oxidation, N-terminal acetylation, and diglycine remnant on lysines as variable modifications.

**Dataset PXD007061.** This dataset has been previously published by Cerletti et al.[4]. The whole proteome turnover was examined in the *H. volcanii* conditional LonB mutant (HVLON3) under reduced (−Trp) and nearly physiological (+Trp) LonB levels. HVLON3 was grown in Hv-Min medium containing ¹⁴NH₄Cl as nitrogen source in absence of Trp and then switched to ¹⁵N-medium with and without Trp (±Lon) to monitor ¹⁵N-label incorporation into newly synthesized proteins over time. In parallel, the degradation of ¹⁴N-labeled proteins was estimated by comparing different time points with an internal standard grown on ¹³C-glucose. Membrane and cytoplasm proteins were prepared and processed by SDS-PAGE, digested with trypsin and analyzed by LC-MS/MS (nanoACQUITY gradient UPLC pump system coupled to an LTQ Orbitrap Elite mass spectrometer). Proteins were originally identified with Sequest embedded in Proteome Discoverer 1.4 searching against the HaloLex *H. volcanii* DS2 proteome (version 24-SEP-2013; https://doi.org/10.5281/zenodo.3565581). Protein turnover as well as statistical analyses were achieved with the online platform QuPE (https://qupe.cebitec.uni-bielefeld.de/QuPE/app). In addition, an in vivo cross-linking assay coupled to immunoprecipitation with α-LonB antibody was performed in the *H. volcanii* H26 wt strain to detect interactions between LonB and its endogenous targets. The quantitative proteomics experiment was performed as a biological triplicate, while the immunoprecipitation was done with four biological replicates.

**Dataset PXD009116.** In this dataset, previously published by Cerletti et al.[43], the proteomes of two halophilic archaea, *H. volcanii* H26 and *N. magadii* ATCC 43099, during exponential and stationary growth were compared. Cultures were grown at 42 °C, shaking at 200 rpm, in rich medium (MGM and Tindall medium, respectively) and membrane and cytoplasm fractions were obtained. Protein samples were processed, digested with trypsin, and subjected to LC-ESI-MS/MS using a nanoACQUITY gradient UPLC pump system (Waters) and an LTQ Orbitrap Elite mass spectrometer. Proteins were originally identified and quantified with MaxQuant (version 1.5.3.17)[83] using the LFQ algorithm searching against the HaloLex *H. volcanii* DS2 proteome (version 24-SEP-2013; https://doi.org/10.5281/zenodo.3565581) and the HaloLex *N. magadii* ATCC 43099 proteome (January 2017; https://doi.org/10.5281/zenodo.3571186; contains 4295 entries, while 4023 were claimed). While *N. magadii* samples were included in the reanalysis for comparative purposes, only the raw data corresponding to samples from *H. volcanii* were used for the combined ArcPP dataset, comprising three and six biological replicates for the cytoplasm and membrane fractions, respectively.

**Dataset PXD010824.** This dataset has been previously published by Abdul Halim et al.[84] and focuses on the analysis of HVO_0405. In order to test if HVO_0405 is a Tat substrate, its twin arginine was mutated to a twin lysine. For cells over-expressing either the WT HVO_0405 or the mutated sequence lacking the twin arginine sequence (both in a Δhvo_0405 background), membrane and cytosolic fractions of cells were isolated. After tryptic digestion, samples were analyzed with a Q Exactive Plus mass spectrometer after chromatographic separation on an UltiMate 3000 RSLCnano system and results were originally searched against the

Uniprot *H. volcanii* DS2 proteome (UP000008243) employing Ursgal and allowing semi-enzymatic cleavage. For each sample, two biological replicates were performed.

**Dataset PXD011012**. This dataset was generated as part of the presented work. The proteome of planktonic and sessile cells at different stages of biofilm development are compared in this dataset. *H. volcanii* H53 liquid cultures were shaken at 250 rpm and grown to an $OD_{600}$ of 0.3. After taking samples, the petri dishes were filled with 10 ml of the culture and incubated statically. After 24, 48, and 72 h, samples were taken from the planktonic phase, the remaining culture was discarded and the sessile cells (biofilm) were washed with 18% (w/v) salt water before scraping off the cells with a razor blade and collecting them in 18% (w/v) salt water. In addition to $OD_{600}$ 0.3, samples from the shaking culture were taken at $OD_{600}$ 0.08 and $OD_{600}$ 0.8. All samples were snap-frozen and stored at −80 °C. Each sample was transferred into 0.5 ml centrifugal filter units (Millipore) and lysed with 400 μl pure $H_2O$ containing protease inhibitors (1 mM PMSF and 1 mM benzamidine). The lysis step was repeated once with $H_2O$ and twice with 2% (w/v) SDS in 10 mM Tris/HCl pH 7.6 containing protease inhibitors as well, in order to solubilize membrane proteins. Proteins were digested separately with Trypsin and GluC using 50 μg each and following the FASP protocol[85], modified according to Esquivel et al.[86]. Multiple, complemental proteases were chosen for increased protein identification and sequence coverage[87]. After digestion, peptides were dried and then labeled with iTRAQ (4plex Applications Kit, AB Sciex) following the manufacturer's protocol. Samples were mixed in combinations that allow for the analysis of proteomic changes over time in the planktonic phase, in the biofilm and between planktonic phase and biofilm.

Mass spectrometric analysis was performed as described[84] with minor modifications. Briefly, samples were desalted on a C18 trap column and peptides were separated on a 50-cm C18 column (2 h gradient, 2–40% (v/v) acetonitrile) directly coupled to a Q Exactive Plus mass spectrometer (Thermo Scientific). MS1 scan parameters were as follows: resolution 70,000, automatic gain control (AGC) target $1 \times 10^6$, maximum IT 50 ms, scan range 375–2000 *m/z*. The top 12 peaks were triggered for HCD fragmentation with a normalized collision energy of 30. MS2 scan parameters were as follows: resolution 17,500, AGC target $1 \times 10^5$, maximum IT 125 ms, fixed first mass 100 *m/z*. A dynamic exclusion list (20 s) was used and charge states 1 and >6 were excluded.

The results were originally analyzed with Ursgal employing the search engines X! Tandem[88], MS-GF+[89], MS Amanda[90] and MSFragger[91]. The database consisted of the UniProt *H. volcanii* DS2 proteome (UP000008243) and the following modifications were included: carbamidomethylation of cysteine (fixed), iTRAQ4plex of any N-terminus (fixed), iTRAQ4plex of tyrosine and lysine (optional), oxidation of methionine (optional). The experiment has been performed as biological triplicates.

**Dataset PXD011015**. In this dataset, previously published by Esquivel et al.[86], the N-glycosylation of pilins and flagellins was characterized. For this purpose, flagellins and pilins were isolated from the supernatant by cesium chloride fractionation. After digestion with GluC, samples were chromatographically separated on an UltiMate 3000 RSLCnano system and analyzed with a Q Exactive Plus mass spectrometer. Two different methods were used: (i) in-source collision-induced dissociation (IS-CID) was applied, leading to the fragmentation of glycans before the MS1 scan, and precursor ions were selected for HCD fragmentation based on mass differences corresponding to monosaccharides; (ii) without IS-CID, the 12 most intense precursor ions were selected for HCD fragmentation. Results were originally analyzed with Proteomatic[92], searching against the Uniprot *H. volcanii* DS2 proteome (UP000008243) and including known *H. volcanii* N-glycans as potential modifications. The H53 wild-type was compared against a knockout strain of the oligosaccharyltransferase AglB and measurements were performed as biological triplicates for both strains.

**Dataset PXD011050**. This dataset, generated as part of this work, was aimed at the characterization of ArtA-dependent protein processing. On the one hand, the dataset used Δ*artA* deletion mutants overexpressing either the wild-type version or site-directed mutants of ArtA in order to determine the active site of ArtA[93]. The plasmids that were transformed into the Δ*artA* deletion mutant AF103 to generate the overexpression strains are listed in Supplementary Data 4. For these strains, the S-layer glycoprotein was purified from the supernatant of exponentially grown cultures by cesium chloride fractionation as described previously[94]. On the other hand, ArtA-dependent processing was compared for H53, Δ*artA*, and Δ*pssA*. The Δ*pssA* mutant FH54 was generated by transforming H53 cells with pFH38 as previously described[95]. In this case, the supernatant and/or membrane fraction of exponentially grown cells have been isolated and used for protease digestion without further fractionation.

All samples were digested with Trypsin and/or GluC following the FASP protocol[85] with minor changes[84,86]. Peptides were reconstituted in 2% (v/v) acetonitrile, 0.1% (v/v) formic acid in $H_2O$ and separated on a C18 PepMap 100 column (15 or 50 cm), coupled to a Q Exactive plus mass spectrometer (Thermo Scientific). MS1 spectra were acquired from 350 to 1600 *m/z* (or 375–2000 *m/z*) at a resolution of 70,000 with an injection time of 50–100 ms and an AGC target of

$1 \times 10^6$ to $3 \times 10^6$. The 12 most intense ions were selected for HCD fragmentation with a normalized collision energy of 27 and fragment ions were analyzed in MS2 at 17,500 resolution, 55–120 ms injection time and $5 \times 10^4$ to $1 \times 10^5$ AGC target. Charge states 1 and >5 were rejected. Results were originally analyzed with Ursgal employing the engines X!Tandem[88], MSFragger[91], and MS-GF+[89] in a search against the Uniprot *H. volcanii* DS2 proteome (UP000008243). Semi-enzymatic cleavage was allowed in order to identify processing sites.

**Dataset PXD011056**. This dataset was previously published by Jevtic et al.[9] and analyzed the proteomic response to environmental stress conditions. The chosen standard conditions refers to growth at 45 °C in Hv-YPC medium with 18% (w/v) salt water and was compared with high and low salt conditions, with 23 and 15% (w/v) salt water, respectively, as well as low and high temperature conditions at 30 and 53 °C, respectively. Total cell extracts were prepared by sonication, solubilization with sodium taurodeoxycholate (0.006% (w/v)) and ultracentrifugation of insoluble material. After digest, a spectral library was generated from pooled peptide aliquots from (i) standard conditions and (ii) all stress conditions. The pooled samples were fractionated into eight fractions by high-pH/reversed-phase separation and each fraction was analyzed using DDA on a TripleTOF 5600+ mass spectrometer after chromatographic separation on an Eksigent nanoLC 425. In addition, for quantitative analyses, unfractionated samples from each condition were analyzed using SWATH acquisition. Protein identification was originally performed using the Paragon search engine v5.0.0.0 implemented in ProteinPilot v5.0 build 4769 against the HaloLex *H. volcanii* DS2 proteome (version 19-NOV-2015; https://doi.org/10.5281/zenodo.3565619). For the reanalysis within the ArcPP, only the samples measured by DDA have been used, i.e., the pooled standards and stress conditions, which have been performed as biological duplicates.

**Dataset PXD011218**. This dataset has been previously published by Costa et al.[7]. To address the impact of the intramembrane protease RhoII on *H. volcanii* physiology, the proteomes of MIG1 (Δ*rhoII*) and the parental H26 strains were compared by shotgun proteomics. Cultures were grown in MGM medium (18% salt water) at 42 °C and samples were taken at exponential and stationary growth phases. Membrane, cytoplasm, and supernatant protein samples were prepared and digested with trypsin. In addition, membrane proteins from exponential phase were fractionated by SDS-PAGE into four sections (PROTOMAP assay). A nanoACQUITY gradient UPLC pump system was used coupled to an LTQ Orbitrap Elite mass spectrometer. Protein identification was originally performed by SEQUEST algorithm embedded in Proteome Discoverer 1.4 searching against the HaloLex *H. volcanii* DS2 proteome (version 24-SEP-2013; https://doi.org/10.5281/zenodo.3565581) The experiment was performed with six biological replicates. This dataset includes files that are part of the dataset PXD009116. In order to avoid duplications, these files were not included here (but only in PXD009116) for the reanalysis.

**Dataset PXD013046**. In this dataset, previously published by Cerletti et al.[5], the impact of the membrane-associated LonB protease on the proteome of *H. volcanii* was examined. To this end, the proteomes of the wild-type strain (H26) and the conditional mutant (HVLON3) with reduced LonB protease levels were compared. As a control, the proteome of strain HVABI, a deletion mutant of the downstream gene *abi*, was analyzed in parallel. These strains were grown in Hv-Min in the absence of Trp and samples were taken for four biological replicates at the exponential and stationary growth phases. Membrane and cytoplasm proteins were prepared, digested with trypsin and analyzed by LC-MS/MS. A nanoACQUITY gradient UPLC pump system was used coupled to an LTQ Orbitrap XL (cytoplasm samples) or a LTQ Orbitrap Elite (membrane samples) mass spectrometer. Protein identification was originally performed by SEQUEST[82] and MS Amanda[90] algorithms embedded in Proteome Discoverer 1.4 searching against the HaloLex *H. volcanii* DS2 proteome (version 24-SEP-2013; https://doi.org/10.5281/zenodo.3565581).

**Dataset PXD014974**. This dataset was generated as part of the presented work. With the aim to analyze the protein translation landscape, whole-cell extracts of H26 cells ($OD_{600}$ of 0.6) were prepared by resuspending snap-frozen cell pellets in 500 μL extraction buffer (150 mM NaCl, 100 mM EDTA, 50 mM Tris pH 8.5, 1 mM $MgCl_2$, 1% (w/v) SDS) and boiling them for 13 min at 95 °C. The cooled-down whole-cell extract was clarified by centrifugation (16,000 × *g* for 10 min), and the clarified supernatants were collected. Proteins were reduced by addition of β-mercaptoethanol (2% (v/v) final concentration) for 1 h in the dark. Proteins were precipitated by addition of acetone (80% (v/v) final concentration) for 1 h at −20 °C. After centrifugation (10 min 16,000 × *g* at 4 °C) the protein pellets were washed with acetone and centrifuged again. The obtained pellets were dissolved at room temperature in 1 ml solubilization buffer (25 mM Tris-HCl, pH 7.1, 6 M urea, 3 M thiourea, 50 mM KCl, 70 mM DTT) and stored overnight at 4 °C. Fifteen micrograms of resolubilized pellet were alkylated with 2-iodoacetamide (30 mM) for 30 min in the dark in a total volume of 110 μl complemented with 50 mM ammonium bicarbonate buffer. Three hundred microliters of urea buffer (8 M urea, 100 mM Tris pH 8.5) and 2 μl 1 M DTT were added and samples were incubated

for 5 min at room temperature. Samples were further processed following the FASP protocol[85]. Digestion was performed overnight at 37 °C using 1 µg trypsin and the digested peptides were eluted with 30 µl of 50 mM ammonium bicarbonate buffer containing 5% (v/v) acetonitrile. Eluates were acidified with 2 µl trifluoroacetic acid.

For analysis of the tryptic peptides, a Q Exactive HF mass spectrometer (Thermo Scientific) coupled to an RSLC system (Ultimate 3000, Dionex, Sunnyvale, CA) was used, similar to ref. [96]. Approximately 1 µg of sample was automatically loaded on the HPLC system, which was equipped with a nano trap column (300-µm inner diameter × 5 mm, packed with Acclaim PepMap 100 C18, 5 µm, 100 Å; LC Packings, Sunnyvale, CA). After 5 min, the peptides were eluted from the trap column and separated using reversed-phase chromatography (Acquity UPLC M-Class HSS T3 Column, 1.8 µm, 75 µm × 250 mm; Waters) using a gradient of 7–27% (v/v) acetonitrile at a flow rate of 250 nl min$^{-1}$ over a period of 90 min, followed by two short gradients of 27–41% (v/v) acetonitrile (15 min) and 41–85% (v/v) acetonitrile (5 min). After 5 min at 85% (v/v) acetonitrile, the gradient was set back to 3% (v/v) acetonitrile over a period of 2 min and allowed to equilibrate for 8 min. All acetonitrile solutions contained 0.1% (v/v) formic acid. Eluting peptides were analyzed in DDA mode which consisted of an MS1 spectrum at a resolution of 60,000 acquired in the Orbitrap ranging from 300 to 1500 $m/z$ with AGC target set to $3 \times 10^6$. From this high-resolution MS scan, the ten most abundant peptide ions were selected for fragmentation if they exceeded an intensity of at least $2 \times 10^4$ counts and if they were at least doubly charged. MS/MS spectra were recorded in the Orbitrap at a resolution of 15,000 with a maximum injection time of 50 ms. The precursor ion isolation window was 1.6 $m/z$. Normalized collision energy was set to 28 and dynamic exclusion was set to 30 s.

The results were originally analyzed using MaxQuant (version 1.6.6.0)[83] using standard parameters and the Uniprot *H. volcanii* DS2 proteome (UP000008243). Two biological replicates were performed.

**General workflow for the reanalysis within the ArcPP**. MS raw data files were downloaded from PRIDE[14] or jPOST[15], converted into the unified HUPO Proteomics Standards Initiative standard file format mzML[97] using the Thermo-RawFileParser (for RAW files from Thermo Scientific)[98] or msConvert (for SCIEX WIFF files, with --filter peakPicking true 1- and --filter zeroSamples removeExtra) included in ProteoWizard[99]. For all subsequent file conversions, all protein database searches, as well as all statistical post-processing (if not indicated otherwise) that were performed within the ArcPP, the Python framework Ursgal (versions 0.6.5 and 0.6.6)[29] has been used. The protein database was derived from the most recent Gold Standard Protein based annotation of the *H. volcanii* genome (version 06-JUN-2019, https://doi.org/10.5281/zenodo.3565631)[100], consisting of 4186 proteins (including 79 spurious annotations and 33 duplicates, all of which were not counted for the final size of the proteome: 4074 proteins). The annotation of this genome (and others from haloarchaea) included extensive efforts to minimize the number of missing protein-coding gene annotations (including small protein-coding genes), e.g., applied algorithms did not include a size cutoff for genes, extensive manual curation was performed[101] and regions not assigned as coding were systematically screened to detect and resolve missing gene calls[102]. The *H. volcanii* database was supplemented with contaminants from the common Repository of Adventitious Proteins (https://www.thegpm.org/crap/). For all proteins, decoys were generated by peptide shuffling, dependent on the protease used for the digest. Protein database searches were then performed against the merged target-decoy database. Results from different search engines were unified within Ursgal, statistically post-processed using Percolator[103] (version 3.4.0) and combined using the combined PEP approach[29,30]. More details, including the initial parameter optimization as well as the combination of multiple datasets are described below.

**Parameter optimization for protein database searches**. For each dataset, protein database searches with X!Tandem[88] have been performed using all combinations of four different precursor mass tolerances (5–20 ppm), five fragment mass tolerances (5, 7.5, 10, 20, 40 ppm for high-resolution MS; 0.1, 0.2, 0.4, and 0.8 Da for low resolution MS) and ten instrument offsets (−10 to 10 ppm). In order to speed up this process, only every second to fifth MS2 spectrum was used for the search. After statistical post-processing, parameter combinations with the highest number of total identified peptides were selected and the best-performing instrument offset was chosen for each MS raw file separately.

**Protein database search for the reanalysis within the ArcPP**. The following protein database search engines, implemented in Ursgal (version 0.6.5 to 0.6.6), were used for all datasets: X!Tandem[88] (version Vengeance), MS-GF+[89] (version 2019.04.18), MSFragger[91] (version 20190222). These search engines were chosen based on their speed and their availability in Ursgal. Besides the precursor and fragment ion mass tolerance and instrument offset determined by parameter optimization (see above), Ursgal's default parameters have been used with the following modifications: oxidation of methionine and N-terminal protein acetylation, both as variable modifications, carbamidomethylation (or methylthio modification, or none, depending on the dataset) of cysteine as fixed modification. For PXD011012, iTRAQ4plex was included as fixed modification of the protein N-terminus and variable modification of lysine and tyrosine. For PXD006877,

Label:13C(5) on proline, Label:13C(6)15N(2) on lysine, and Label:13C(6) on arginine were included as variable modifications. A maximum of two and three missed cleavages was allowed for datasets using Trypsin and GluC as protease, respectively. If samples were fractionated, results from one engine for all fractions were merged before statistical post-processing. Results from multiple search engines were afterward combined using the combined PEP approach[29] and filtered by a combined PEP ≤ 1%. In case of discrepant identifications for the same spectrum by different database search engines, results were sanitized. To this end, the best PSM for each spectrum was chosen if there was no ambiguity or if the best PSM had a combined PEP that was an order of magnitude better than other identifications. Otherwise, all PSMs for that spectrum were rejected.

**Comparison with original search results**. Results from the original analysis were obtained from PRIDE, jPOST or provided by the individual research groups. This also applies to datasets that have not been published previously; they had been analyzed (as described above) independently of the ArcPP by the corresponding research group. In order to allow for a fair comparison, original search results were filtered by PEP ≤ 1% and sanitized as well. Furthermore, peptides smaller than six or larger than 50 amino acids were excluded. Finally, modifications other than the ones included in the reanalysis were removed as well.

**Protein inference and calculation of peptide and protein FDR**. The most recent annotation of the *H. volcanii* genome contained 19 sequences that had one or more identical duplicates. In total, 52 sequences were merged into 19 new protein names by randomly choosing one of the corresponding HVO IDs as representative and indicating the number of duplicates for each group. Besides this removal of identical sequences, identified peptide sequences that are part of multiple proteins were handled by a simplistic protein inference model, since their number is relatively small in *H. volcanii*. Non-proteotypic peptides were assigned to one protein if, out of the group of proteins that contain this peptide, only one protein was identified by other peptides in the same sample. Otherwise, the identification was kept as a protein group. Protein groups mapping on multiple proteins identified by other peptides were not taken into account for further analysis (total protein number, etc.). Protein groups not mapping onto any other protein were regarded as a single protein for further analyses.

Peptide and protein FDRs were calculated for each dataset separately as well as for the combination of all datasets. In both cases, the picked protein FDR approach[32] was used similar to Wang et al.[21]. On the peptide level, the best (lowest) Bayes PEP (from the combined PEP function in Ursgal) for each peptide sequence was chosen. After sorting, the list was traversed from top to bottom and the cumulative number of decoys was divided by the number of cumulative targets, yielding an empirical $q$-value. A second traversal from bottom to top, changing $q$-value from the first traversal to the minimum $q$-value observed so far, ensured monotonicity. For the estimation of protein FDRs, a score for each protein was calculated as the sum of $-\log_{10}$ transformed minimal Bayes PEPs from all identified sequences of that protein. Only peptide sequences with a peptide FDR ≤ 1% were taken into account. The protein scores were sorted, and $q$-values were calculated by traversing the list from top to bottom and bottom to top, as done for peptide $q$-values.

Peptides and proteins were regarded as confidently identified if their corresponding FDR was smaller than, or equal to, 1% and 0.5%, respectively. In addition, they were required to be supported by at least two PSMs. The effects of this filtering are described in Fig. S1 and the elimination of all decoy peptide hits with a peptide FDR ≤1% highlights the usefulness of this approach. We rejected the commonly used threshold to require two identified peptides because that interferes with identification of smaller proteins and has previously been shown to not be suitable for distinction between correct identifications and false positives[104].

It should be noted that calculations of peptide and protein FDRs have been performed for the combination of all datasets as well as for each individual dataset separately. The number of identifications reported for individual datasets, as well as for the comparison between datasets (Fig. 4) correspond to dataset-specific FDR calculations, while the overall identifications correspond to the FDR calculations for the combination of all datasets. This leads to a small number of proteins being identified only in the combined dataset but not in any individual dataset and vice versa.

**MW, pI and hydrophobicity calculation**. Molecular weight, pI, and hydrophobicity were computed by custom PERL scripts. For molecular weight, mono-isotopic masses were used (as, e.g., listed in Expasy (https://web.expasy.org/findmod/findmod_masses.html#AA)). Computation of pI values is based on the pK values for amino acids at internal, N-terminal, and C-terminal positions[105]. For hydrophobicity, the GRAVY index was computed, based on the hydropathy index of amino acids[106].

**Prediction of signal peptides and transmembrane domains**. The *H. volcanii* proteome was analyzed using TMHMM 2.0[107] for TM domains, SignalP 5.0[48] (organism group: archaea) for predictions of the Sec pathway, FlaFind[46] for predictions of pilins, processed by SPIII (PibD), TatFind[45] for Tat substrates, LipoP 1.0[108] for lipobox-containing proteins, and TatLipo[47] for Tat substrates containing

a lipobox, which involves cleavage by an as of yet unidentified bacterial SPII analog. Using these predictions, each protein was assigned to a single category based on positive predictions in a sequential decision tree as follows: TatLipo (Tat (lipobox)) → LipoP (Sec (lipobox)) → TatFind (Tat (SPI)) → FlaFind (Pil (SPIII)) → SignalP (Sec (SPI)) → TMHMM (TM) → Cyt. Proteins with at least two TM domains are considered integral membrane proteins, while proteins with one TM domain were categorized into TM N-term and TM C-term if their TM domain was within the first and last 50 amino acids, respectively. For some analyses (Fig. 3c, d), the categories Tat (lipobox), Sec (lipobox), Tat (SPI), Pil (SPIII), and Sec (SPI) were summarized as secreted proteins.

**Semi-enzymatic protein database search**. Protein database search for semi-enzymatic peptide has been performed using the same workflow as described above with the following exceptions. The Ursgal parameter semi_enzyme has been set to True. Furthermore, before statistically post-processing the results with Percolator, PSMs were grouped based on fully enzymatic and semi-enzymatic peptides and PEP calculations were performed for each group separately. This grouped validation approach results in more accurate FDRs on PSM level[109]. Results were merged and peptide and protein FDRs were calculated as described above. Since the increased search space in a semi-enzymatic search can nevertheless lead to higher FDRs, the results from this search were not taken into account for the final number of identified proteins and peptides, but were only used for the comparison with signal peptide prediction engines. Furthermore, samples digested with GluC were excluded from the comparison, because a high number of semi-enzymatic peptides was identified, indicating a reduced site specificity of the enzyme. Results from immunoprecipitations and PXD000202 were excluded as well. In addition, for increased confidence, a minimum of five PSMs was required for the identification of semi-tryptic peptides. Finally, proteins with more semi- than fully-tryptic peptides were not taken into account, since increased proteolytic cleavage instead of a defined signal peptide cleavage was assumed.

Results were compared with predictions for Sec (SPI), Tat (SPI), and Sec (SPII) processing from SignalP 5.0, because it was shown to be the only prediction engine to accurately predict this variety of signal peptide CS in archaea[48]. Since SignalP 5.0 has not been trained on Tat substrates containing a lipobox, results from TatLipo[47] were used to override Tat (SPI) predictions from SignalP 5.0 with Tat (lipobox). If a semi-tryptic peptide starting at the predicted CS was identified, the predictions was regarded as correct. If a semi-tryptic peptide starts within a range of plus/minus three amino acids, the predicted CS was refined. If both cases were not fulfilled but a fully enzymatic peptide was identified starting at least three amino acids N-terminal of the predicted CS, the prediction was regarded as incorrect. Proteins, for which tryptic cleavage sites around the predicted CS prevented theoretical peptides with a length of 5–50 amino acids, or for which an N-terminal lipid modification was predicted, were counted but not classified as correct/incorrect, since an identification of semi-tryptic peptides for the predicted CS would not be possible through the employed methods.

**Genomic islands with low protein identification rates**. The analysis was performed separately for each replicon. Proteins were ordered serially along the replicon, based on the start of the coding region (which corresponds to the N-terminus for proteins encoded on the forward strand and to the C-terminus for proteins encoded on the reverse strand). For each gene, the corresponding protein identification rate was computed, considering 25 genes on each side, thus covering 51 genes. The circularity of all replicons was taken into account. Identification rates were in the range of 14 (27.5%) to 48 (94.1%). Closely spaced genes with a low identification rate (up to 20 identifications, 39.2%) are reported as low identification islands. Two islands with low identification rates correspond to prophages according to PhySpy[110,111].

**Statistical analysis of arCOG classes**. For three groups ((i) proteins present in all seven whole proteome datasets, (ii) proteins only identified in one whole proteome dataset, (iii) proteins not identified within the ArcPP), the distribution of arCOG classes[57] was analyzed in comparison to their background distribution within the whole *H. volcanii* proteome. Significance was evaluated using Fisher's exact test, considering for each group of proteins: (a) the number of identified proteins that belong to an arCOG category and (b) the number of identified proteins which do not belong to that arCOG category; equivalent numbers (within arCOG category; outside arCOG category) are computed for the background (whole theoretical proteome). A Bonferroni correction for multiple testing was applied on resulting *p*-values.

**Urease activity assay**. *H. volcanii* H26 was grown in GMM (Hv-Min medium with 20 mM glycerol as the carbon source and 10 mM NH$_4$Cl as the nitrogen source) or CM (ATCC974 medium composed of 2.14 mM NaCl, 246 mM MgCl$_2$, 28.7 mM K$_2$SO$_4$, 0.9 mM CaCl$_2$, 0.5% tryptone (Bacto$^{TM}$) and 0.5% yeast extract (Oxoid$^{TM}$), adjusted to pH 6.8 with 1 M KOH). Cells were grown in 50 ml cultures (in 250 ml Erlenmeyer baffled flasks) at 42 °C with rotary shaking at 200 rpm.

Urease activity was monitored by detection of NH$_4^+$ production by the phenol-hypochlorite method as previously described[112] with the following modifications. Cells were harvested in log phase (OD$_{600}$ of 0.3–0.6) by centrifugation (F14-6x250

LE rotor, 2500 × *g*, 5 min, room temperature). Cell pellets (8 OD$_{600}$ units total) were washed with 10 ml of buffer A (20 mM Tris-HCl buffer pH 7.2 supplemented with 2 M NaCl) by similar centrifugation. Cell pellets were resuspended to a final volume of 0.2–0.25 ml in buffer A and transferred to a 1.8 ml microfuge tube on ice. Samples were mixed with disruptor beads (0.2 g, 0.1 mm diameter, Chemglass) and vortexed (5 × 1 min with 2 min breaks on ice). Samples were centrifuged at 12,500 × *g* for 5 min at 4 °C and the cell lysate supernatant was transferred to a new 1.8-ml tube on ice. The protein concentration of the cell lysate was determined by Bradford Assay (BioRad) with NaOH included as 20 µl of 0.1 N NaOH stock per 200 µl assay to facilitate protein solubility. Bovine serum albumin (BSA) was used as the protein standard. Cell lysate (1.5–5 mg protein per ml) was used for the urease assay. Reactions (75 µl final volume), consisting of 65 µl cell lysate and 10 µl of 10% urea (w/v) in buffer A or 10 µl buffer A for the background control, were incubated at 25, 37, 42, 60, and 80 °C. Aliquots (10–15 µl) of the reaction were removed over time (0, 1 h, 2 h, and 3 h) and immediately assayed for NH$_4^+$ by the phenol-hypochlorite method[112] using (NH$_4$)$_2$SO$_4$ as the standard.

**Reporting summary**. Further information on research design is available in the Nature Research Reporting Summary linked to this article.

## Data availability
The data that support this works are available from the corresponding author upon reasonable request. The raw files of all new proteomic datasets are available on PRIDE with the following identifiers: PXD011050, PXD011012, and PXD014974. The annotated proteome of *H. volcanii* is deposited at https://doi.org/10.5281/zenodo.3565580. PSMs and summarized result files for all datasets are deposited at https://doi.org/10.5281/zenodo.3825856. Furthermore, all main result files and all meta data is available at https://github.com/arcpp/ArcPP. The source data underlying Figs. 1, 2a-c, 3a-d, 4a, b, and Supplementary Figs 1a-d, 2a, b, 3a-c, 4a-f, 5 and 6d are provided as a Source data file.

## Code availability
Only freely available software has been used as described in the Methods. Analysis scripts that allow reproduction of the results are available at https://github.com/arcpp/ArcPP.

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

## Acknowledgements

The helpful discussions and valuable developments in Ursgal by Johannes Leufken and Manuel Kösters are greatly appreciated. Jonathan Stoltzfus and Ronald Rodriguez are greatly acknowledged for assisting in strain and sample generation. We also thank Dr. Uli Ohmayer from PolyQuant GmbH for performing sample preparation and LC/MS-MS data acquisition of PXD014974. S.S. was supported by the German Research Foundation (DFG Postdoctoral Fellowship, 398625447). M.P. was supported by the National Science Foundation Grant 1817518. Work in A.M.'s laboratory was funded by the German Research Foundation (Grant MA1538/24-1) in the frame of SPP2002. J.M.-F. received funding from the U.S. Department of Energy, Physical Biosciences Program (DOE DE-FG02-05ER15650) the National Institutes of Health (NIH R01 GM57498) and the National Science Foundation NSF MCB-1642283. S.F.-C. was supported by intramural funding from the department of Biochemistry III House of the Ribosome, by the German Research Foundation (DFG): individual research grant (FE1622/2-1), and collaborative research center SFB/CRC 960 (SFB960-B13). R.D.C. received funding from the National Agency for the Promotion of Science and Technology-ANPCyT- (PICT1477) and the MINCyT-BMBF (Argentina-Germany) (AL/13/02) awarded to R.D.C. and A.P. M.H. acknowledges support by the German Research Foundation (DFG, HI737/12-1).

## Author contributions

S.S., Z.A., M.C., R.D.C., S.F.-C., M.I.G., M.H., C.L., A.M., J.M.-F., R.A.P., F.P., A.P., H.U., and M.P. contributed to the sharing and organization of datasets. S.S., Z.J., R.K., and G.L. were involved in MS sample preparation. S.S. performed the reanalysis of the datasets. Results were analyzed and interpreted by S.S., M.C., R.D.C., S.F.-C., M.I.G., C.L., A.M., J.M.-F., R.A.P., F.P., and M.P. The web database was developed by C.F. with contributions by S.S. Urease activity assays were performed by J.M.-F. S.S. conceived the idea together with M.P., who supervised the project. The paper was written by S.S. with contributions of Z.A., M.C., R.D.C., S.F.-C., M.I.G., C.L., A.M., J.M.-F., F.P., A.P., and M.P. All authors have given approval to the final version of the paper.

## Competing interests

The authors declare no competing interests.
