## [Peer Review File · Nature Communications]

Reviewers' comments:

Reviewer #1 (Remarks to the Author):

This manuscript describes the Archaeal Proteome Project (ArcPP), a community-based effort focused on the comprehensive analysis of archaeal proteomes. Starting with the model archaeon *Haloferax volcanii*, the authors have reanalyzed MS datasets from various strains and culture conditions, and identified over 72% of the reference proteome with an median protein sequence coverage of 56%. This expanded dataset enabled a deeper inspection of *H. volcanii* cell biology, leading to new insights into processes such as terminal protein maturation, N-glycosylation, protein degradation and metabolism.

In general, this paper is scientifically sound and well-written. The content is logical, systematic, and straight-forward. As such, this report represents a large amount of concerted work and thus should be significant interest to the scientific community. While the authors are not the first to propose and execute such an idea to combine multi-related datasets for expanded analyses, this appears to be the first for archaeal systems.

Some minor comments are listed below:

1. The author might consider a slightly revised version of the title (especially the second part) to better deliver a meaningful “science nugget about what is learned” to engage the lay reader.

2. Line 41 – this general approach (combine datasets to achieve greater proteome coverage/identifications has been used with other systems (especially bacteria) and should be more adequately referenced here.

3. Line 65 – the MS techniques highlighted in Fig 1 have dramatically different dynamic range measurement capabilities, which will affect the depth of proteome measurement and the ability to detect low abundance components. The authors should better clarify this issue and how it would impact their results.

4. Line 87 – it might be worth some value for the authors to expand slightly on why use 3 different search algorithms, and why choose the specific 3 they did?

5. Line 118 – FDR control on large datasets is a formidable issue – the authors should expand their discussion a bit more here to clarify this point and how confident they are in their chosen approach.

6. Line 136 – here or elsewhere it might be worth mentioning metrics of protein overlap between samples (maybe include Venn diagrams as supplemental?). Readers will be interested not only in proteins that are unique to individual datasets but also the range and degree of numerical overlap between samples (or even all samples).

7. Fig 4a – the x-axis label is confusing and should be clarified.

8. Line 278 – there are two exciting implications of this work – a deeper dataset for enhancing reannotation (proteogenomics) and the identification/ functional assignment of “proteins of unknown function.” Both are briefly mentioned here, but this text should be significantly expanded to better highlight and communicate this work. It would be helpful to include some tables of proteins of unknown function that were highly associated with certain (or all) strains.

Reviewer #2 (Remarks to the Author):

ArcPP ms review

In “ The Archaeal Proteome Project – advancing knowledge about archaeal cell biology through comprehensive proteomics “, Schulze and colleagues present ArcPP, a proteomics database for archaea. They collate datasets from a variety of strains and growth conditions across five labs, 12 datasets in the research community, and use updated or novel bioinformatics tools to re-analyze the spectra. They have detected 72% of the proteome. They make several important advancements based on these re-analyses, including cell surface N-glycosylation sites, protein degradation, etc. Particularly useful are their suggestions for improving experimental design for future proteomics experiments. This study therefore represents an advance in both the fields of archaeal molecular biology and microbial proteomics. They have provided a valuable resource for the archaeal research community. Statistical analyses are comprehensive and appropriate for the data presented. I have some concerns and questions that are intended to improve and clarify the manuscript:

Abstract

- Revise the phrase “already lead to...”. Does this mean that these discoveries resulted from previous work, or were they newly discovered here?

Introduction

The authors state that “comparable community efforts for prokaryotes have been lacking thus far...” Can the authors give additional context for how their study relates to the field of proteomics in bacteria and archaea? For example, a Peptide Atlas for *Hbt. salinarum*, a related halophile, was published in 2008, and recent studies report large datasets for bacterial pathogens (e.g. <https://www.nature.com/articles/s41598-017-10059-w>). Please refer to relevant studies in this field here and/or elsewhere in the paper where relevant (e.g. conclusion). I think at least a few more sentences in the introduction along these lines would help clarify the significance of the advance of the study for the general reader.

Results

1. I know some of the proteomics datasets analyzed have been published and some have not, but the reader really needs to dig in the paper to figure out which is which. Can the authors clarify this (perhaps in the methods by simply putting an inline citation next to the PXD number, or by adding a column to the supplementary table that lists the PRIDE accessions for each dataset)? Related to this, one of the major claims of the paper is that improvements on the initial analysis of the data have been made in terms of FDR and PSM, etc, but what is the reference point if the dataset was not previously published? Were the data analyzed only once previously as described in the methods? This is alluded to in the methods here and there, but perhaps this could be clarified by more specific language there or elsewhere as appropriate?
2. The authors make several discoveries regarding certain protein properties, but recapitulated previous knowledge here as well. Can the authors clarify which is which throughout? In addition, for the broader audience, can the authors contextualize the importance / biological relevance of certain protein properties (e.g. this was done for N-terminal protein processing, but not for the biological function of acetylation of cleaved methionine, RhoII, urease, etc.)? Where the function is not known, please state this to clarify how this study provides insight in these aspects.
3. Regarding commonly identified proteins, can the authors quantify the statement “For most of these proteins deletion mutants could not be generated...” (line 212). How do these protein functions compare with the essential functions found in other archaea via Tn-seq methods (PMID: 23487778; 30464174)? Are essential functions conserved?

4. Fig 4b and extended data Fig 5: when the authors are talking about “enrichment” of commonly identified proteins in arCOG functional categories, have they conducted a hypergeometric test to determine if these enrichments are significant (e.g. using Fisher’s Exact Test or hypergeometric test) relative to the number of proteins predicted for that category in the genomic background?

5. Regarding small proteins that weren’t identified (line 263 and elsewhere): identification depends on the genome annotation used for spectral matching? As the authors are likely aware, annotation algorithms have a size cutoff for coding ORFs. Has the annotation at the Zendo accession been revised to include small proteins? Can the authors clarify this in the text where appropriate? The authors note that the ArcPP can help with improved genome annotation – perhaps they could note small proteins as a specific example?

6. Related to the point above, the authors have placed their discussion of identification of eight proteins previously annotated as “nonfunctional” in extended data, but this is an important point and shows that the current study has made strides toward improved annotation. If space allows, can the authors bring this analysis into the main text?

7. It is appreciated how all of the datasets, analysis code, and relevant accessions are freely available, promoting data provenance. However, the main repository for this study is on github with csv flat files. On github, while the authors freely admit in their notes on github the limitations of distributing the data this way (E.g. not searchable), I wonder if it’s possible to mount a rudimentary searchable database for the community (many of whom are unfamiliar with coding and how to use github) to use the data? Or if a database is outside the scope of the current study, perhaps the authors could provide some additional instructions on the main readme on the github repository for how the readers can download, parse and search the file themselves (e.g. by upload to various analysis tools or softwares)?

Minor comments

Some minor typos and grammatical errors throughout, please correct.

Reviewer #3 (Remarks to the Author):

The authors have assembled a re-analysis of available *Haloferax volcanii* mass spectrometry-based proteome experiments to generate a thorough analysis of the detected proteome of this model archaea. This appears to be the deepest coverage proteomic analysis of any archaeon, which is

notable. The authors also identify characteristics of individual studies that improve different aspects of the proteomic analyses. This will be of utility for designing future studies.

While these are valuable contributions in their own right, interest in them is likely restricted to those working directly in the sub-field of microbial proteomics. My major suggestion for the authors is that they re-focus or re-organize the manuscript to more clearly articulate the importance of biological conclusions that can be drawn from this deep analysis. What have we learned about archaeal cell and systems biology through this effort and why is it important? The authors mention several interesting biological conclusions, but in my view, none are fully described or supported. I applaud the author's efforts in re-analysis here, but to make the case that this is a worthwhile approach that can be used to study any prokaryote, I think the authors need to spend more effort describing the importance and novelty of, and support for, their biological conclusions.

I found the MS data re-analysis approach and procedure to be well-described and well-documented. I have no major concerns about the MS-data processing approaches employed here and it appears to me that these analyses are reproducible given the information provided in the reviewed documents and the associated GitHub page.

Another concern is the discussion of "...differential protein regulation dependent on culture conditions." (line 65-66). When this is discussed, there is little acknowledgment that there are differences in proteome data acquisition between experiments, other than culture conditions, that could lead to differences in protein identification. For instance, in lines 251-260, the authors describe that urease subunits were only detected in the dataset PXD006877 and the authors suggest that this is because this was the only experiment conducted using minimal media. This is also the only experiment that used SCX for peptide fractionation and the only one applying SILAC for quant. The authors may well be correct in their assertion that this differential detection is due to the culture conditions, but they cannot rule out other factors. Therefore, I suggest the authors soften the language that they use to describe what can be learned by examining detection patterns across experiments as they do here. I argue that its not reasonable to discuss 'differential protein regulation' between experiments conducted in such different ways and that 'differential protein detection' may be more appropriate.

Something else to consider: how different are the strains used here from type strain/ genome strain, and are these differences in genome content possible factor driving protein identification rates or differences in "islands" with reduced detection?

Minor:

Mio is an atypical abbreviation for Million- please spell out the full word.

Reviewer #1 (Remarks to the Author):

This manuscript describes the Archaeal Proteome Project (ArcPP), a community-based effort focused on the comprehensive analysis of archaeal proteomes. Starting with the model archaeon *Haloferax volcanii*, the authors have reanalyzed MS datasets from various strains and culture conditions, and identified over 72% of the reference proteome with a median protein sequence coverage of 56%. This expanded dataset enabled a deeper inspection of *H. volcanii* cell biology, leading to new insights into processes such as terminal protein maturation, N-glycosylation, protein degradation and metabolism.

In general, this paper is scientifically sound and well-written. The content is logical, systematic, and straight-forward. As such, this report represents a large amount of concerted work and thus should be of significant interest to the scientific community. While the authors are not the first to propose and execute such an idea to combine multi-related datasets for expanded analyses, this appears to be the first for archaeal systems.

Some minor comments are listed below:

1. The author might consider a slightly revised version of the title (especially the second part) to better deliver a meaningful “science nugget about what is learned” to engage the lay reader.

Since our findings cover multiple areas of archaeal cell biology, we found it hard to choose a specific finding for the title and conveying several of them within 15 words seemed impossible. Therefore, we kept the original title and the lay reader is referred to the abstract for a summary of more specific results.

2. Line 41 – this general approach (combine datasets to achieve greater proteome coverage/identifications) has been used with other systems (especially bacteria) and should be more adequately referenced here.

We extended references on other papers describing the combination of multi-related datasets for expanded analyses of prokaryotic proteomes. However, we would like to highlight that, while indeed the “idea to combine multi-related datasets for expanded analyses” has been proposed before, ArcPP significantly differs from these approaches in the scale, the community focus (including extensibility) and the biological information obtained. We now more clearly lay out those differences in the introduction and discussion.

3. Line 65 – the MS techniques highlighted in Fig 1 have dramatically different dynamic range measurement capabilities, which will affect the depth of proteome measurement and the ability to detect low abundance components. The authors should better clarify this issue and how it would impact their results.

We agree with the reviewer that differences in sample preparation and MS data acquisition will impact the identification of proteins; we have stated this more clearly in the manuscript and also changed

“differential protein regulation” to “differential protein identification”, as suggested by reviewer 3. However, our general results in regard to the depth of proteome coverage and the comparison of different sample processing and measurement approaches are not affected by this. Furthermore, we think that comparisons between datasets can still provide valuable insights and testable hypotheses for the roles of proteins under specific conditions. This is especially true for the frequent detection of proteins with common physicochemical properties, an aspect that is stated now as well.

4. Line 87 – it might be worth some value for the authors to expand slightly on why use 3 different search algorithms, and why choose the specific 3 they did?

We expanded slightly on the combined PEP approach, which combines results from multiple search engines by taking advantage of the increased confidence in PSMs commonly identified by different search engines. This approach had been shown to increase the number of PSMs while estimating PEPs more accurately than e.g. simply merging the results.

We added, in the corresponding methods section, that the specific three search engines were chosen based on their speed and their availability in Ursgal. We also note in that context that an expansion to even more search engines did not lead to improved results.

5. Line 118 – FDR control on large datasets is a formidable issue – the authors should expand their discussion a bit more here to clarify this point and how confident they are in their chosen approach.

Since FDR control on large datasets is discussed more extensively in recent publications (as referenced to in the manuscript) we focused on presenting the data (plots for FDR distributions on peptide and protein level, including different filtering thresholds) in a way that allows the reader to get a clear picture of the applied approaches and estimated FDRs. We stressed that our analysis approaches were aimed at providing highly confident results.

6. Line 136 – here or elsewhere it might be worth mentioning metrics of protein overlap between samples (maybe include Venn diagrams as supplemental?). Readers will be interested not only in proteins that are unique to individual datasets but also the range and degree of numerical overlap between samples (or even all samples).

The overlap between samples was summarized and presented in Fig. 4a as the number of proteins that were identified in 1, 2, ..., all 7 of the whole proteome datasets. Unfortunately, a Venn diagram for 7 datasets is not practicable/readable, but in order to allow readers to explore these overlaps in more detail, csv tables for proteins identified in 1, 2, ..., 7 datasets are provided on GitHub and a search/filtering for proteins identified in a specific dataset, or in the overlap between datasets, is possible through the web database that is provided now.

7. Fig 4a – the x-axis label is confusing and should be clarified.

The x-axis title of Fig. 4a has been changed to “Number of datasets sharing protein identifications” and the figure legend has been rephrased for clarification.

8. Line 278 – there are two exciting implications of this work – a deeper dataset for enhancing reannotation (proteogenomics) and the identification/ functional assignment of “proteins of unknown function.” Both are briefly mentioned here, but this text should be significantly expanded to better highlight and communicate this work. It would be helpful to include some tables of proteins of unknown function that were highly associated with certain (or all) strains.

As explained below (reviewer 2, comment 6), we think that the more detailed discussion on the identification of proteins previously annotated as “nonfunctional” proteins is better suited for the Supplementary Notes. However, we extended the main text on genomic islands with low protein identification rates.

We agree with the reviewer that an improved functional assignment of proteins in *H. volcanii* is important and that our results provide first hints towards potential functional roles of various proteins of unknown function. As mentioned above (reviewer 1, comment 6), the provided tables and web database allow for straight-forward exploration of the data, including the analysis of overlaps between datasets or proteins uniquely identified in one dataset as well as the filtering for proteins of unknown function. However, more detailed analyses of proteins associated with different strains requires the quantitative analysis of the ArcPP datasets, which, due to the different labeling and quantification methods, is out of scope of the current manuscript.

Reviewer #2 (Remarks to the Author):

ArcPP ms review

In “ The Archaeal Proteome Project – advancing knowledge about archaeal cell biology through comprehensive proteomics “, Schulze and colleagues present ArcPP, a proteomics database for archaea. They collate datasets from a variety of strains and growth conditions cross five labs, 12 datasets in the research community, and use updated or novel bioinformatics tools to re-analyze the spectra. They have detected 72% of the proteome. They make several important advancements based on these re-analyses, including cell surface N-glycosylation sites, protein degradation, etc. Particularly useful are their suggestions for improving experimental design for future proteomics experiments. This study therefore represents an advance in both the fields of archaeal molecular biology and microbial proteomics. They have provided a valuable resource for the archaeal research community. Statistical analyses are comprehensive and appropriate for the data presented.

I have some concerns and questions that are intended to improve and clarify the manuscript:

Abstract

- Revise the phrase “already lead to...”. Does this mean that these discoveries resulted from previous work, or were they newly discovered here?

The phrase has been changed to “provided new insights into ...”.

Introduction

The authors state that “comparable community efforts for prokaryotes have been lacking thus far...” Can the authors give additional context for how their study relates to the field of proteomics in bacteria and archaea? For example, a Peptide Atlas for *Hbt. salinarum*, a related halophile, was published in 2008, and recent studies report large datasets for bacterial pathogens (e.g. <https://www.nature.com/articles/s41598-017-10059-w>). Please refer to relevant studies in this field here and/or elsewhere in the paper where relevant (e.g. conclusion). I think at least a few more sentences in the introduction along these lines would help clarify the significance of the advance of the study for the general reader.

We significantly extended the introduction on different large-scale proteomics approaches in bacteria and archaea and their limitations. We now more clearly lay out how ArcPP significantly distinguishes itself from these approaches in the scale, the community focus (including extensibility) and the biological information obtained.

Results

1. I know some of the proteomics datasets analyzed have been published and some have not, but the reader really needs to dig in the paper to figure out which is which. Can the authors clarify this (perhaps in the methods by simply putting an inline citation next to the PXD number, or by adding a column to the supplementary table that lists the PRIDE accessions for each dataset)? Related to this, one of the major claims of the paper is that improvements on the initial analysis of the data have been made in terms of FDR and PSM, etc, but what is the reference point if the dataset was not previously published? Were the data analyzed only once previously as described in the methods? This is alluded to in the methods here and there, but perhaps this could be clarified by more specific language there or elsewhere as appropriate?

The literature reference or unpublished status of the datasets has been stated more prominently in the methods. PMIDs have been added to Supplementary Table 5. It should be noted that references were also already included in Fig. 1, i.e. with the first mention of the PRIDE ID.

It is now explicitly stated in the methods that unpublished datasets have been analyzed independently of the ArcPP by the corresponding research group, using bioinformatic tools as described for each individual dataset (each dataset description contains information about the “original analysis/peptide identification” that was performed).

2. The authors make several discoveries regarding certain protein properties, but recapitulated previous knowledge here as well. Can the authors clarify which is which throughout? In addition, for the broader audience, can the authors contextualize the importance / biological relevance of certain protein properties (e.g. this was done for N-terminal protein processing, but not for the biological function of acetylation of cleaved methionine, RhoII, urease, etc.)? Where the function is not known, please state this to clarify how this study provides insight in these aspects.

Throughout the manuscript, we have clarified which previous findings were confirmed or significantly extended through our large-scale analyses within the ArcPP and which new insights were gained through our data. Furthermore, we extended various parts on the context and biological relevance of our results.

3. Regarding commonly identified proteins, can the authors quantify the statement “For most of these proteins deletion mutants could not be generated...” (line 212). How do these protein functions compare with the essential functions found in other archaea via Tn-seq methods (PMID: 23487778; 30464174)? Are essential functions conserved?

We have rephrased this paragraph to include quantitative statements in regard to *H. volcanii* deletion mutants as well as a comparison to homologs for essential genes found by transposon tagging (TnSeq data) in *Sulfolobus islandicus*.

4. Fig 4b and extended data Fig 5: when the authors are talking about “enrichment” of commonly identified proteins in arCOG functional categories, have they conducted a hypergeometric test to determine if these

enrichments are significant (e.g. using Fisher's Exact Test or hypergeometric test) relative to the number of proteins predicted for that category in the genomic background?

We have now statistically analyzed the distribution of different arCOG categories for the set of proteins identified in all, one, or none of the datasets compared to their distribution in the whole theoretical proteome using Fisher's exact test and applying Bonferroni correction for multiple testing. Significant results are indicated in Extended Data Fig. 5 and confirm our previous statements.

5. Regarding small proteins that weren't identified (line 263 and elsewhere): identification depends on the genome annotation used for spectral matching? As the authors are likely aware, annotation algorithms have a size cutoff for coding ORFs. Has the annotation at the Zendo accession been revised to include small proteins? Can the authors clarify this in the text where appropriate? The authors note that the ArcPP can help with improved genome annotation – perhaps they could note small proteins as a specific example?

We have now added a sentence to the methods section to more clearly state the careful analysis to minimize the number of missing gene calls, especially for small protein-coding genes. Concerning the reviewer's question, as detailed in e.g. PMID:26042526 and PMID:18592220, reliability to the genome annotation was a key issue for more than one decade of work, and this included efforts to reduce missing gene calls of small protein-coding genes. Moreover, modern gene callers no longer apply a size cutoff but use sophisticated statistical analyses instead, thus reducing the number of missing gene calls. Furthermore, as detailed in e.g. PMID:17444671, PMID:16950390, and PMID:18592220, GC-rich genomes (and nearly all haloarchaeal genomes are GC-rich) lead to ORF overprediction, including spurious N-terminal extensions, leading to artificially extended ORFs, which then are more likely to exceed a potential size cutoff. In a procedure developed while working on PMID:275193433, all intergenic regions were subjected to BLASTx analysis against all proteins in the UniProt database. Since the genus *Haloferax* has genome sequences for a plethora of species, it is likely that small (or otherwise missing) genes would have been called in at least one of them, and would therefore have been detected in the PMID:275193433 related analysis. Taken together, we are confident to say that any possible care has been taken to minimize the number of missing gene calls for small proteins.

We strongly agree that proteogenomic will hopefully even further enhance genome annotation with respect to calling of genes for small proteins. Likewise, we hope that proteogenomic analyses will help to further reduce start codon assignment errors. However, after careful consideration, we think that small proteins are not a well suited example for the ArcPP-driven improvement of the genome annotation, because the low identification rate for small proteins, as noted in the manuscript, indicate that dedicated sample processing methods are required for the comprehensive identification of this subproteome. Only when respective datasets become available, this could be used for an improved genome annotation in regard to small proteins.

6. Related to the point above, the authors have placed their discussion of identification of eight proteins previously annotated as “nonfunctional” in extended data, but this is an important point and shows that the

current study has made strides toward improved annotation. If space allows, can the authors bring this analysis into the main text?

We have now moved parts of the section on low identification islands to the main text but we think that the “nonfunctional” proteins are more suited for the Supplementary Notes, since their biological relevance is limited. There are 11 “identified nonfunctional proteins”. (a) One is not nonfunctional but a genome sequence error. That case is biologically not interesting. Of course, the issue needs to be resolved in the database (yet pending). (b) A second one is not nonfunctional but “technically tagged” as nonfunctional because the gene traverses the point of ring opening of a plasmid, which disconnects the two parts of the coding region. Again, this case is biologically not interesting. (c) Four proteins have been identified, but all identified peptides are upstream of the point of gene disruption. That indicates that the transcription and translation signals remain intact and that these are not “pseudogenes” but “disrupted genes”. The protein fragments need to be stable enough to be detectable by proteomics, but that does not imply that either one of them is also functional. The one case where a function had been assigned (PMID:25904898) was later attributed to an overinterpretation of low sequence coverage data (PMID:29495512). Thus, we question that this is relevant enough for the main text. (d) in four cases, the identifications are on the edge of the FDR threshold, and only one of them made it into the final list of identified proteins. Stringent filtering, as applied by us, will drastically reduce false positive but will not completely eliminate them. Those four “identified” nonfunctional proteins might belong to that unavoidable set of false positives and should therefore not be overinterpreted. (e) The final case is a protein which was identified even though it lacks a canonical start codon (ATG, GTG, TTG) between the upstream in-frame stop codon and the first identified peptide. In principle, this case would have been interesting to subject to a more detailed analysis of the available proteomic data. However, several Arg and Glu residues are unfavorably positioned in the relevant region and make an eventual N-terminal peptide unidentifiable. Thus, also this case cannot be advanced to become biologically interesting. Taken together, the “nonfunctional issue” seems not relevant enough for the main text.

7. It is appreciated how all of the datasets, analysis code, and relevant accessions are freely available, promoting data provenance. However, the main repository for this study is on github with csv flat files. On github, while the authors freely admit in their notes on github the limitations of distributing the data this way (E.g. not searchable), I wonder if it’s possible to mount a rudimentary searchable database for the community (many of whom are unfamiliar with coding and how to use github) to use the data? Or if a database is outside the scope of the current study, perhaps the authors could provide some additional instructions on the main readme on the github repository for how the readers can download, parse and search the file themselves (e.g. by upload to various analysis tools or softwares)?

We have created a central webpage (www.archaealproteomeproject.org will go online with the acceptance of the manuscript, reviewer access is provided through: https://www.sas.upenn.edu/~sschulze/arcpp_test/arcpp.html) as landing page for central information and links to the GitHub repository (with updated readme), download pages for all relevant data files as well as a web database that allows for a straight-forward exploration of identified proteins and peptides, as well as their overlap between datasets, arCOG categories, etc.

Minor comments

Some minor typos and grammatical errors throughout, please correct.

We have again carefully read the manuscript and hope to have caught all typos and grammatical errors.

Reviewer #3 (Remarks to the Author):

The authors have assembled a re-analysis of available *Haloferax volcanii* mass spectrometry-based proteome experiments to generate a thorough analysis of the detected proteome of this model archaea. This appears to be the deepest coverage proteomic analysis of any archaeon, which is notable. The authors also identify characteristics of individual studies that improve different aspects of the proteomic analyses. This will be of utility for designing future studies.

While these are valuable contributions in their own right, interest in them is likely restricted to those working directly in the sub-field of microbial proteomics. My major suggestion for the authors is that they re-focus or re-organize the manuscript to more clearly articulate the importance of biological conclusions that can be drawn from this deep analysis. What have we learned about archaeal cell and systems biology through this effort and why is it important? The authors mention several interesting biological conclusions, but in my view, none are fully described or supported. I applaud the author's efforts in re-analysis here, but to make the case that this is a worthwhile approach that can be used to study any prokaryote, I think the authors need to spend more effort describing the importance and novelty of, and support for, their biological conclusions.

Clarifications throughout the manuscript, thanks to all three reviewers' insightful comments, now more clearly lay out the significance of this community-based proteomics effort to the microbiology community beyond the sub-field of microbial proteomics. Additional *in vivo* and *in silico* data now included in the manuscript (e.g. Supplementary Fig. 6, Supplementary Note 3, also see answers to the other reviewers' comments) support the importance of the biological conclusions and also serve as examples underscoring this broader impact of the manuscript.

I found the MS data re-analysis approach and procedure to be well-described and well-documented. I have no major concerns about the MS-data processing approaches employed here and it appears to me that these analyses are reproducible given the information provided in the reviewed documents and the associated GitHub page.

Another concern is the discussion of "...differential protein regulation dependent on culture conditions." (line 65-66). When this is discussed, there is little acknowledgment that there are differences in proteome data acquisition between experiments, other than culture conditions, that could lead to differences in protein identification. For instance, in lines 251-260, the authors describe that urease subunits were only detected in the dataset PXD006877 and the authors suggest that this is because this was the only experiment conducted using minimal media. This is also the only experiment that used SCX for peptide fractionation and the only one applying SILAC for quant. The authors may well be correct in their assertion that this differential detection is due to the culture conditions, but they cannot rule out other factors. Therefore, I suggest the authors soften the language that they use to describe what can be learned by examining detection patterns across experiments as they do here. I argue that its not reasonable to discuss 'differential protein

regulation' between experiments conducted in such different ways and that 'differential protein detection' may be more appropriate.

We agree with the reviewer that differences in sample preparation and MS data acquisition will impact the identification of proteins; we have stated this more clearly in the manuscript and also changed "differential protein regulation" to "differential protein identification". However, we think that comparisons between datasets can still provide valuable insights and testable hypotheses for the roles of proteins under specific conditions. This is especially true for the frequent detection of proteins with common physicochemical properties, an aspect that is stated now as well.

With respect to the urease subunits: the physicochemical properties of these proteins (size, pI, hydrophobicity) do not suggest difficulties in their identification. Thus, we would not expect that SCX fractionation or SILAC would strongly influence the identification of corresponding peptides. Together with the fact that multiple subunits were only identified in GMM, we considered it likely that indeed the presence/absence of glycerol as carbon source is the distinguishing feature making those proteins identifiable only in the respective dataset. Consistent with this hypothesis we are excited to report that follow-up in vivo urease activity assays confirm that urease activity is only detected in GMM but not in CM cultures (Supplementary Fig. 6)

Something else to consider: how different are the strains used here from type strain/ genome strain, and are these differences in genome content possible factors driving protein identification rates or differences in "islands" with reduced detection?

As described in Supplementary Note 6, all used strains are a direct offspring of the wildtype isolate *Haloferax volcanii* strain DS2, which also is the type strain of that species and the strain that has been used for genome sequencing. All used strains are cured of a small plasmid (pHV2) (parts of which are used in cloning vectors, which led to identification of some of them). Specific genes were deleted by a markerless gene deletion approach (pop-in, pop-out). Somehow, one of the plasmids has managed to get integrated into the chromosome while converting the wildtype to the laboratory strains. Nevertheless, all these strains are as isogenic to the genome-sequenced type strain as possible. We do not see any reason to attribute protein identification rates to strain differences.

Minor:

Mio is an atypical abbreviation for Million- please spell out the full word.

Mio. has been removed and the suffix M was used for numbers on the axis in order to be consistent with other figures.

REVIEWERS' COMMENTS:

Reviewer #2 (Remarks to the Author):

The authors have adequately addressed all of my previous comments. Particularly commendable among the author responses is the mounting of a website with a searchable database of the data - a resource that I think will be particularly useful for the research community. I have no further critiques.